# CAUSALITY-INSPIRED ROBUSTNESS FOR NONLINEAR MODELS VIA REPRESENTATION LEARNING

## ABSTRACT

Distributional robustness is a central goal of prediction algorithms due to the prevalent distribution shifts in real-world data. The prediction model aims to minimize the worst-case risk among a class of distributions, a.k.a., an uncertainty set. Causality provides a modeling framework with a rigorous robustness guarantee in the above sense, where the uncertainty set is data-driven rather than pre-specified as in traditional distributional robustness optimization. However, current causality-inspired robustness methods possess finite-radius robustness guarantees only in the linear settings, where the causal relationships among the covariates and the response are linear. In this work, we propose a nonlinear method under a causal framework by incorporating recent developments in identifiable representation learning and establish a distributional robustness guarantee. To our best knowledge, this is the first causality-inspired robustness method with such a finite-radius robustness guarantee in nonlinear settings. Empirical validation of the theoretical findings is conducted on both synthetic data and real-world single-cell and ICU data, also illustrating that finite-radius robustness is crucial.

## 1 INTRODUCTION

In real-world applications, data distributions often shift between training and deployment environments, leading to degraded performance of machine learning (ML) models. These shifts can arise from changes in data collection methods, environmental conditions, or adversarial perturbations. Distributional robustness addresses this challenge by ensuring that models perform well across a range of possible distributions, rather than just the training distribution. This is particularly critical in high-stakes domains such as healthcare, finance, and autonomous systems, where unreliable predictions can have severe consequences. By focusing on robustness, we aim to build models that generalize reliably under some distribution shifts. The goal of distributional robustness, specifically, is to optimize models for the worst-case scenario within a predefined set of possible distributions, known as the uncertainty set. This approach contrasts with traditional empirical risk minimization (ERM), which focuses solely on average performance on training data. By minimizing the worst-case risk among an uncertainty set, distributionally robust models are better equipped to handle unseen distributional shifts. The uncertainty set can be defined in various ways, such as through statistical distances (Mohajerin Esfahani & Kuhn, 2018), moment constraints (Wiesemann et al., 2014; Bertsimas et al., 2018; Hanasusanto & Kuhn, 2018), or causal assumptions (Rojas-Carulla et al., 2018; Rothenhäusler et al., 2021). This flexibility allows the framework to be tailored to specific application domains and types of distribution shifts.

Causal models offer a principled way to define uncertainty sets based on the underlying data-generating process, rather than relying on postulated distances. Based on causal relationships, these models typically try to identify invariant features that remain stable in different environments (Peters et al., 2016; Louizos et al., 2017; Pfister et al., 2021), providing a natural foundation for robustness. This data-driven approach avoids the

need for ad hoc definitions of uncertainty sets, which may not capture the true nature of distributional shifts. Furthermore, causal frameworks enable the incorporation of domain knowledge, enhancing the interpretability and reliability of the resulting models.

Most causality-inspired robustness methods enforce invariance constraints, for instance, approaches such as Arjovsky et al. (2019); Christiansen et al. (2021) aim to ensure that predictions remain invariant under arbitrarily strong perturbations or distribution shifts. In contrast, methods like anchor regression (Rothenhäusler et al., 2021) and DRIG (Shen et al., 2023) offer *finite-radius robustness* guarantees for an uncertainty set of finite-strength perturbations. However, existing methods that provide finite-radius guarantees have been confined to settings where the relationships among variables are linear, restricting their applicability to domains such as certain economic or physical systems. Yet, many modern applications, including image recognition and genomics, exhibit highly nonlinear interactions, and extending finite-radius robustness guarantees to such settings remains an open challenge.

To our best knowledge, we make the first attempt to develop a causality-inspired robustness method with a finite-radius robustness guarantee in nonlinear settings. This is achieved by integrating causal principles with modern techniques in representation learning, allowing us to handle nonlinear dependencies while maintaining robustness. This advancement opens up new possibilities for applying causal methods to a broader range of problems, including those involving high-dimensional and nonlinear data. Unlike traditional approaches such as adversarial training, and predefined distributional robustness optimization (DRO) frameworks, which often fail to generalize effectively in the presence of complex dependencies or multi-source heterogeneity, our method provides a unified solution that bridges DRO, causality, and nonlinear representation learning. This new proposal addresses the limitations of existing methods and offers a more comprehensive and flexible framework for robust prediction. Furthermore, our framework avoids relying on strong assumptions about causal identifiability or predefined robustness metrics, instead allowing the data itself to drive the learning process. This flexibility is particularly important in real-world scenarios, where the training data do not contain enough information to identify the underlying causal mechanisms, but its heterogeneity can be exploited to define the uncertainty set. We validate our theoretical results through experiments on both synthetic datasets and real-world single-cell, and ICU datasets.

## 2 RELATED WORK

### 2.1 DISTRIBUTIONALLY ROBUST OPIMIZATION.

Despite significant advances in machine learning, the deployment of ML models in real-world applications often reveals their limitations and vulnerabilities. These challenges arise from distributional shifts (Mansour et al., 2009; Rothenhäusler et al., 2021; Shen et al., 2023), adversarial attacks (Goodfellow et al., 2014; Madry et al., 2017), and noisy or incomplete data.

This led to an increase in interest in combining robust prediction schemes with other methodologies, such as representation learning and causality (Schölkopf et al., 2021), or reinforcement learning and distributional robustness (Smirnova et al., 2019; Lu et al., 2024). One foundational work toward achieving robustness in a distributional sense is Delage & Ye (2010) who formalized DRO using moment-based uncertainty sets. A more recent approach considers DRO in terms of Wasserstein distance (Mohajerin Esfahani & Kuhn, 2018; Hanasusanto & Kuhn, 2018; Kuhn et al., 2019). An interesting remark on the Wasserstein approach is noted in Gao et al. (2024), essentially relating LASSO and several other estimators to solutions of DRO problems. However, though it enjoys certain theoretical guarantees, the approach considers the worst-case distribution contained within a region in Wasserstein distance, hence yielding overly conservative predictions. A different yet similar approach can be seen in Popescu (2005) or Zhen et al. (2025), where structural properties of distributions (e.g. symmetry, unimodality, and convexity) are integrated into an uncertainty set based on moments. Wiesemann et al. (2014) generalizes moment-based ambiguity sets using conic

inequalities, allowing for more flexible moment constraints beyond just first and second moments. For further background in general DRO, we refer the reader to Rahimian & Mehrotra (2022).

## 2.2 INVARIANCE-FOCUSED FRAMEWORKS

Traditional approaches to robustness, such as regularization or adversarial training, provide valuable insights but often struggle to generalize across different types of challenges. This limitation has driven interest in methods that aim to exploit the invariant properties of the data. Arjovsky et al. (2019) propose Invariant Risk Minimization (IRM), a framework designed to encourage models to learn invariant predictors across diverse training environments. This paradigm is especially relevant when models are deployed in scenarios where distributional shifts are expected, as it focuses on isolating relationships that remain robust under varying conditions.

Risk extrapolation (REx) (Krueger et al., 2021) refines the principle of invariance by explicitly penalizing risk disparities across training domains. Its V-REx variant minimizes the variance of domain-specific risks through a regularization penalty. This formulation encourages robustness against extreme shifts by extrapolating risks beyond observed domains. Although V-REx theoretically recovers causal predictors under mechanism shifts, its empirical performance is highly sensitive to label noise and spurious correlations in finite-sample regimes. Crucially, V-REx inherits IRM's limitation of requiring strict risk invariance—an assumption violated when heterogeneous environments contain non-invariant causal mechanisms. Adaptive risk minimization (ARM) (Zhang et al., 2021) represents a paradigm shift toward test-time adaptability. Unlike static multi-environment frameworks (e.g., IRM, V-REx), ARM meta-trains models to dynamically adjust parameters using context sets from test distributions.

Other directions of theoretical studies in this direction include studying the objective of IRM, and its potential problems (Rosenfeld et al., 2020), as well as distributional matching guarantees in terms of environments needed (Chen et al., 2022). Further interesting results on online domain generalization are considered in Rosenfeld et al. (2022).

**Causality-inspired methods.** In contrast to classical DRO methods, where the set of perturbations against which the model is protected is prespecified and often overly conservative, recent causality-inspired frameworks focus on *isolating* the relevant directions along which distributional shifts occur, in a data-driven way. A novel work by Rothenhäusler et al. (2021) introduces a causality inspired framework that guarantees robustness against shifts in mean. Building on this foundation, Shen et al. (2023) proposed the *Distributional Robustness via Invariant Gradients* (DRIG) framework, which extends these guarantees to both mean and variance. Both works exploit the heterogeneity of the data originating from multiple sources.

## 2.3 LATENT REPRESENTATION LEARNING.

One of the central concerns in context of representation learning and dimensionality reduction is the question of identifiability — whether the features or representations learned genuinely reflect the underlying data structure or are they not more than products of the chosen combination of hyperparameters. Research in this field has advanced in many interesting directions and has become increasingly productive in recent years (Khemakhem et al., 2020a;b; Schölkopf et al., 2021; Kivva et al., 2022; Moran et al., 2021; Wang et al., 2023). This leap forward features an examination of the assumptions regarding latent variables, the nature of their generative processes and distributions, and even of their decoder functions. Notable work by Khemakhem et al (2020a) investigates the setting of conditionally independent latent variables given an observed auxiliary variable, through the iVAE framework.

Other approaches have sought to provide identifiability guarantees using polynomial decoders (Ahuja et al., 2023), volume-preserving decoders (Yang et al., 2022), sparse VAEs (Moran et al., 2021), and about

identifiability in general (Roeder et al., 2021; Buchholz et al., 2023). However, despite all these impressive results, a breakthrough work by Kivva et al. (2022) sets a new standard and challenges the necessity of auxiliary information in the latent structure for achieving strong guarantees for a broad class of functions. Most of these works conclude that identifiability of the hidden representation is possible only up to an affine transformation. Saengkyongam et al. (2023) propose the Rep4Ex framework, which uses interventional heterogeneity to recover latent state representations up to an affine transformation, enabling reliable extrapolation to unseen interventions. Assuming a linear structural causal model with exogenous interventions and full-support residuals, Rep4Ex enforces "linear invariance" through a custom autoencoder objective, learning representations that generalize across observed and off-support actions. In contrast, our approach frames robustness to bounded shifts as a finite-radius DRO problem, providing closed-form certificates under noise assumptions.

## 3 METHOD

### 3.1 MODEL SETUP

We observe a $d$-dimensional covariate $X$ and a real target variable $Y$. It is often reasonable to assume the data distribution is entailed by an underlying causal mechanism, which is much weaker than assuming the identifiability of such a causal mechanism. We would like to also account for nonlinear causal relationships. One way to incorporate nonlinearity is through a nonlinear representation map of the covariates $X$ that transforms a complex distribution of $X$ to a better-structured latent space where the causal relationship can be as simple as linear.

Specifically, we assume there exists a function $\phi^* : \mathbb{R}^d \to \mathbb{R}^k$ for $d \geq k$ such that the observed variables $(X, Y)$ follow a structural causal model (SCM) in the usual or observational setting:

$$\begin{pmatrix} \phi^*(X) \\ Y \end{pmatrix} = \mathbf{B} \begin{pmatrix} \phi^*(X) \\ Y \end{pmatrix} + \varepsilon \tag{1}$$

where $\varepsilon \in \mathbb{R}^{k+1}$ has zero expectation, and is allowed to have correlated components and $\mathbf{B}$ denotes the adjacency matrix of the causal graph. We will refer to $\phi^*$ as *true encoder*. We assume $\mathbf{I} - \mathbf{B}$ to be invertible, where $\mathbf{I}$ denotes the identity matrix, which is guaranteed if the graph is acyclic. This model allows for nonlinear causal relationships between the covariates and between $X$ and $Y$, which can be represented as a linear SCM up to a nonlinear transformation of the covariates.

In practice, we often encounter distribution shifts due to interventions on observed or latent variables. We consider a multi-environment setup as in Shen et al. (2023), where for each environment indexed by $e$, the distributions of the transformed covariates and the response are shifted by a random, additive intervention, i.e.,

$$\begin{pmatrix} \phi^*(X^e) \\ Y^e \end{pmatrix} = \mathbf{B} \begin{pmatrix} \phi^*(X^e) \\ Y^e \end{pmatrix} + \varepsilon + \delta^e \tag{2}$$

where $\delta^e \in \mathbb{R}^{k+1}$, denoting the additive intervention, is independent of $\varepsilon$. We assume during training, we have access to multiple environments $\mathcal{E}$, where one of them, indexed by $0 \in \mathcal{E}$, is the observational setting such that $\delta^0 = 0$, while the others are interventional settings, each with a distinct intervention variable $\delta^e$.

Furthermore, we are interested in out-of-distribution prediction, where the data may exhibit a different underlying distribution, namely, according to

$$\begin{pmatrix} \phi^*(X^v) \\ Y^v \end{pmatrix} = \mathbf{B} \begin{pmatrix} \phi^*(X^v) \\ Y^v \end{pmatrix} + \varepsilon + v \tag{3}$$

where the intervention variable $v \in \mathbb{R}^{k+1}$ follows an unseen distribution different from that of $\delta^e$'s and is independent of $\varepsilon$, while the transformation $\phi^*$ and the graph structure $\mathbf{B}$ stay the same.

Our target is an optimal nonlinear prediction model that is robust among distributions generated according to equation 3 for a class of new interventions. When $\phi^*$ is the identity map, Shen et al. (2023) presented an approach to achieve an optimal linear model that is robust among certain test distributions. This motivates us to develop a two-step approach that consists of a first representation learning step to learn $\phi^*$ up to a certain non-identifiable equivalence class and a second step applying a robust prediction objective on top of the learned representations. For an overview of notation we refer to table 2.

### 3.2 TWO-STEP APPROACH

#### 3.2.1 REPRESENTATION LEARNING STEP

Shen & Meinshausen (2024) proposed Distributional Principal Autoencoder (DPA) that learns low-dimensional representations while preserving the data distribution in the reconstructions. We build our representation learning step upon DPA, where the encoder maps from the data space to the latent space and the stochastic decoder maps from the latent space to the data space:

$$ X \xrightarrow{\phi(.)} Z \xrightarrow{\text{dec}(.,\tilde{\varepsilon})} \widehat{X} $$

where $\tilde{\varepsilon}$ follows the standard normal distribution, and where $\phi$ and dec denote an encoder and decoder, respectively. The original DPA ensures that the decoder produces reconstructions $\hat{X}$ that follow the same distribution as the original data $X$, and the encoder minimizes the unexplained variability in the conditional distribution of $X|\phi(X)$. These are achieved by minimizing the following objective function jointly over the encoder and decoder:

$$ L_{\text{DPA}} = \mathbb{E}_X \mathbb{E}_{\tilde{\varepsilon}} \|X - \text{dec}(\phi(X), \tilde{\varepsilon})\| - \frac{1}{2}\mathbb{E}_X \mathbb{E}_{\tilde{\varepsilon},\tilde{\varepsilon}'} \|\text{dec}(\phi(X), \tilde{\varepsilon}) - \text{dec}(\phi(X), \tilde{\varepsilon}')\|, $$

where $\tilde{\varepsilon}, \tilde{\varepsilon}'$ are independently drawn from the standard normal distribution. For a fixed encoder, the objective function $L_{\text{DPA}}$ is the expected negative *energy score* for the conditional distribution of $X|\phi(X)$; see Shen & Meinshausen (2024) for details.

Here, we account for heterogeneity across different environments by encouraging the learned representations from the encoder to follow a mixture of Gaussians. To ensure this, a third neural network, called the *prior* network $g$, is introduced in addition to the standard encoder-decoder framework already present. The prior network takes the environment labels $E$ as input and produces a sample of the latent vector $Z_g$ from a mixture of Gaussians; specifically, $g(E, \xi) = \mu_g(E) + \xi\Sigma_g(E)$ for $\xi$ standard normal, and $\mu_g$ and $\Sigma_g$ denote the estimated mean and root of the covariance matrix, respetively.

$$ E \xrightarrow{g(.,\xi)} Z_g $$

Furthermore, since we want to encourage the latent to emulate a mixture of Gaussians, we also augment the loss function of the DPA to enforce the encoder output matches the distribution of the prior. The new loss term is the negative energy score for the conditional distributions of $\text{enc}(X)|E$:

$$ L_G = \mathbb{E}_{X,E} \mathbb{E}_\xi \|\phi(X) - g(E, \xi)\| - \frac{1}{2}\mathbb{E}_E \mathbb{E}_{\xi,\xi'} \|g(E, \xi) - g(E, \xi')\| $$

where $\xi, \xi'$ are independently drawn from a standard normal distribution. The formulation can be thought of as a conditional version of the DPA. Indeed, just as the optimum of the loss function $L_{\text{DPA}}$ in Shen & Meinshausen (2024) was motivated by the goal of ensuring

$$ \text{dec}^{\text{opt}}(z, \tilde{\varepsilon}) \quad = \quad X|\{\phi^{\text{opt}}(X) = z\}, \quad \text{in distribution} \quad \forall z, $$

the augmentation loss function $L_G$ was inspired by the goal of achieving

$$g^{\text{opt}}(e, \xi) \quad = \quad \phi^{\text{opt}}(X)|\{E = e\}, \quad \text{in distribution} \ \ \forall e$$

where $\phi^{\text{opt}}$, $\text{dec}^{\text{opt}}$, and $g^{\text{opt}}$, denote the optimized encoder, decoder and prior network, respectively.

The final augmented loss function reads

$$L_{\text{RL}} = L_{\text{DPA}} + \alpha L_G, \tag{4}$$

for a selected hyperparameter $\alpha$. We define

$$(\phi^{\text{opt}}, \text{dec}^{\text{opt}}) \in \underset{(\phi, \text{dec})}{\arg\min} \, L_{\text{RL}},$$

where we also sometimes refer to $\phi^{\text{opt}}$ as $\widehat{\phi}$. Moreover, optimizing $L_{\text{RL}}$ gives $\widehat{\phi}(X) = \widehat{Z}$. Since DPA learns the distribution of $X$, as well as the distribution of its principal components in the latent space, the estimated latent vector is an affine transformation of the true latent vector $\widehat{z} = Az + c$, for an invertible matrix $A$ and a vector $c$. Affine identifiability is guaranteed in a range of settings, and it can be achieved by imposing conditions on the distribution of latent, variables (for example, presence of interventions), or on the mixing function. Since the autoencoder scheme learns to match the distribution of $X$, affine identification is ensured by Lemma 2 from Kivva et al. (2022).

It is worth noting that there are also other similar results which can be used to ensure affine identification in this setup similar to this. For example, since interventions are present in the considered setting, Lemma 5 from Ahuja et al. (2023) can also be applied using VAE (Kingma et al., 2013). These approaches mainly differ in the assumptions they impose. Assumptions in Kivva et al. (2022) are distributional, they posit a GMM structure in the latent space and require matching of distributions, whereas assumptions in Ahuja et al. (2023) constrain the decoder class (e.g., to polynomials) and require an exact reconstruction identity (which can be enforced via a VAE). Both frameworks are valid in this methodology under their respective assumptions; we adopt the distributional setting as in Kivva et al. (2022) because it naturally complements the second step of our method. Some known applicable results on identifiability can be found in Appendix B.

### 3.2.2 CAUSALITY-INSPIRED ROBUSTNESS STEP

Motivated by its guarantees for distributional robustness in linear settings and its adjustable robustness radius parameter, $\gamma \geq 0$, we employ the DRIG method (Shen et al., 2023) to learn a robust linear model on top of the representations learned in the first step. Specifically, let $\widehat{Z}^e = \widehat{\phi}(X^e)$ be the learned representations for each environment $e \in \mathcal{E}$. Additionally, DRIG requires that all environments be centered relative to the mean of the reference environment, for both $\widehat{Z}^e, Y^e$. To satisfy this requirement, we adopt the following centering step;

$$\widehat{Z}^e_c = \widehat{Z}^e - \mathbb{E}[\widehat{Z}^0] = A(Z^e - \mathbb{E}[Z^0]) = AZ^e_c, \ \ Y^e_c = Y^e - \mathbb{E}[Y^0]$$

where this operation constitutes a linear transformation of the true latent variable. We then define the linear coefficients by

$$\widehat{b}_\gamma = \underset{b}{\arg\min} \, L^\gamma_{\text{CIR}}(b), \tag{5}$$

for

$$L^\gamma_{\text{CIR}}(b) = \mathbb{E}[Y^0_c - b^\top \widehat{Z}^0_c]^2 + \gamma \sum_{e \in \mathbb{E}} \omega^e \left( \mathbb{E}[Y^e_c - b^\top \widehat{Z}^e_c]^2 - \mathbb{E}[Y^0_c - b^\top \widehat{Z}^0_c]^2 \right),$$

where $\omega^e > 0$ denote environment weights and it holds $\sum_{e \in \mathcal{E}} \omega^e = 1$. For example, in the uniform case, $\omega = 1/|\mathcal{E}|$, or $\omega^e = n_e/n$, where $n_e$ denotes the number of samples from environment $e$. We remark that $\widehat{b}_\gamma$

can be computed explicitly, by solving the equations obtained by first order conditions, and using empirical risk instead of population risk;

$$\widehat{b}_\gamma = \left((1-\gamma)\widehat{\mathbf{Z}}_c^{0\top}\widehat{\mathbf{Z}}_c^0 + \gamma\sum_{e\in\mathcal{E}}\omega^e\widehat{\mathbf{Z}}_c^{e\top}\widehat{\mathbf{Z}}_c^e\right)^{-1}\left((1-\gamma)\widehat{\mathbf{Z}}_c^{0\top}\mathbf{Y}_c^0 + \gamma\sum_{e\in\mathcal{E}}\omega^e\widehat{\mathbf{Z}}_c^{e\top}\mathbf{Y}_c^0\right), \tag{6}$$

where $\mathbf{Z}, \mathbf{Y}$ are in bold since they represent matrices and vectors of of dimension *sample size× latent dimension*, *sample size×*1, respectively.

### 3.2.3 FINAL ALGORITHM

We now summarize our two-step algorithm as follows:

- Learn $\widehat{\phi}$ by optimizing $L_{\text{RL}}$ in equation 4.

- Estimate $\widehat{b}$ from $(\widehat{Z}_c, Y_c)$ as the solution of equation 6.

- Define the final prediction model as $\widehat{f}(x) = \widehat{b}^\top \widehat{z}_c$, where $\widehat{z}_c = \widehat{\phi}_c(x)$.

We call our method CIRRL (Causality-Inspired Robustness via Representation Learning).

### 3.3 THEORETICAL GUARANTEES

This subsection formalizes the theoretical guarantees for robustness and identifiability. Proofs of all the results can be found in Appendix C. To quantify the robustness of a prediction model $f$, we use the following worst-case risk:

$$\mathcal{L}_\gamma(f) = \sup_{v\in C^\gamma} \mathbb{E}_v[Y - f(X)]^2,$$

where

$$C^\gamma = \left\{v\in\mathbb{R}^{k+1} \mid \mathbb{E}[vv^\top] \preceq S^0 + \gamma\sum_{e\in\mathbb{E}}\omega^e\left(S^e - S^0 + \mu^e\mu^{e\top}\right)\right\}$$

with $S^e = \text{cov}[\delta^e]$ and $\mu^e = \mathbb{E}[\delta^e]$. In the case where $f$ can be rewritten as $f(x) = b^\top\phi(x)$ for some vector $b$ and an arbitrary function $\phi$, the worst-case risk $\mathcal{L}_\gamma(f)$ can be rewritten as $\mathcal{L}_\gamma(b^\top\phi) = \sup_{v\in C^\gamma}\mathbb{E}_v[Y - b^\top\phi(X)]^2$. The following lemma characterizes the set of perturbations against which the model is robust, in terms of $v$ and its distribution. Let $\phi_c(X) := \phi(X) - \mathbb{E}\phi(X^0)$ denote the centered version of $\phi$.

**Proposition 1.** *Assume that the SCMs (Equations 2, 3) hold as described above. Let $\phi : \mathbb{R}^d \to \mathbb{R}^k$ be an affine transform of $\phi^*$, i.e. $\phi(x) = N\phi^*(x) + m$ for an invertible $k \times k$ matrix $N$, and a vector $m$. Then, the loss function $\mathcal{L}_\gamma(b^\top\phi_c)$ has explicit form, namely*

$$\mathcal{L}_\gamma(b^\top\phi_c) = \mathbb{E}[Y^0 - b^\top\phi_c(X^0)]^2 + \gamma\sum_{e\in\mathbb{E}}\omega^e\left(\mathbb{E}[Y^e - b^\top\phi_c(X^e)]^2 - \mathbb{E}[Y^0 - b^\top\phi_c(X^0)]^2\right),$$

*where $\omega^e > 0$ denote environment weights such that $\sum_{e\in\mathbb{E}}\omega^e = 1$.*

This proposition provides a characterization of robustness guarantees, showing that the model can tolerate perturbations $v$ whose second moments are bounded by a weighted combination of second moments of training environments. Importantly, the degree of robustness can be controlled through the hyperparameter $\gamma$. Under relatively mild conditions (see Appendix B), the latent variables can be identified up to an affine transformation. For example, for a piecewise affine decoder function $f$.

**Lemma 2.** *[Kivva et al. (2022)] Let $E$ denote an auxiliary variable (e.g. environment label) and $\dim(E) = 1$. Under the Gaussian mixture model (GMM) for $(E, Z)$[1], as defined in Appendix B and for $f$ a piecewise affine decoder function, if $f$ is weakly injective (see Appendix B), $P(E, Z)$ can be identified from $P(X)$ up to an affine transform.*

Since the optimized DPA matches the distribution of $\widehat{X}$ to the distribution of $X$ we obtain an affine transform of the true latents. This provides a crucial foundation for designing a method that leverages a linear estimator on top, to obtain predictions. The following technical assumption is the last ingredient needed to ensure optimality within all $\mathbb{L}^2$ functions.

**Assumption 1.** *Let $\mathbb{E}[v] = \Sigma[(\boldsymbol{I} - \boldsymbol{B})_{1:k,\cdot}^{-1}]^\top \alpha$ for some $\alpha \in \mathbb{R}^k$, where $\Sigma = \mathrm{Cov}[\varepsilon + v]$.*

This assumption means that *the average effect of $v$ on $Y$ in the SCM passes only through $Z$*. This is easy to see, as the resulting is a vector of linear combinations of rows of $(\boldsymbol{I} - \boldsymbol{B})_{1:k,\cdot}^{-1}$, which corresponds to the total causal effect on the vector $Z$. For a more detailed discussion of technical assumptions, see Appendix C.2. Based on the above results, the following theorem establishes the central theoretical finding of this work. It indicates that our learned prediction model is the most robust among all square-integrable functions.

**Theorem 3.** *Under SCMs described in equations 2, 3, and Assumption 1 assume that $\varepsilon, v$ are elliptical (Definition 2) and recall that $\varepsilon$ and $v$ are independent. Let $\mathcal{X}$ denote the unbounded support of $X$ and $\mathbb{L}^2(\mathcal{X})$ the usual space of square-integrable functions over $\mathcal{X}$. Then,*

$$\mathcal{L}_\gamma(\widehat{f}) = \min_{f \in \mathbb{L}^2(\mathcal{X})} \mathcal{L}_\gamma(f)$$

*for $\widehat{f}(X) = \widehat{b}^\top \widehat{\phi}_c(X)$.*

*Remark* 1. Examples of elliptical distributions include multivariate Gaussian and multivariate $t$-distributions.

## 4 EXPERIMENTS

We validate our theoretical results in a simulated environment and two real datasets. For context, we compare our method to other approaches that can be adopted in the setting concerning nonlinear and robust prediction. Namely, empirical risk minimization (ERM), invariant risk minimization (IRM), risk extrapolation (V-REx), adaptive risk minizimation (ARM), and DRIG estimator. All of IRM, ERM, V-REx, ARM, and CIRRL are parametrized by fully connected neural networks with a comparable number of parameters, while DRIG is applied naively to observed $X$ values. A single (L4) GPU was used to train all models, and Adam (Kingma & Ba, 2014) was used for optimization. For details of the hyperparameters used in optimization, we refer to Appendix D. To conserve space, dataset descriptions are provided in Appendix E.

**Artificial data.** We exclude DRIG from simulation studies since it shows far more inferior performance than other models. It is clear that CIRRL significantly outperforms all considered baselines (Figure 1). In addition to the usual setting, we also evaluate CIRRL in a misspecified scenario through disregarding the distributional assumption in Theorem 3. Specifically, we generate the OOD set from a $\chi^2$-distribution[2]. The rest of the experimental setup mirrors that of the scenario in the well-specified case.

**Real data.** CIRRL delivers substantial and reliable gains under real-world distribution shifts (Table 1). On the ICU benchmark, CIRRL achieves the lowest mean squared error and the smallest worst-environment loss; while DRIG is close on some metrics, CIRRL is favored when the median is taken into account, indicating

---

[1]More concretely, $Z|E = e$ should follow a Gaussian with a mean and covariance dependent on $e$.

[2]Its degrees of freedom approximately corresponding to $1-$norm of $v$

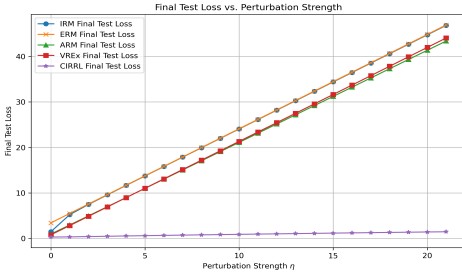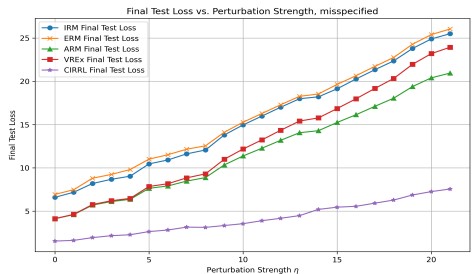

Figure 1: Synthetic dataset for well-specified (left) & misspecified setting (right)- final OOD MSEs for IRM, CIRRL, ERM, ARM, VREx for various perturbation strengths $\eta$ (Appendix E). The inferiority of IRM to other methods could be attributed to the *finite* nature of the perturbations (see Appendix F.1).

a consistently better central tendency as well as improved tail behavior. Crucially, both CIRRL and DRIG dramatically reduce the catastrophic worst-case errors exhibited by ERM, this demonstrates that invariant regularization visibly mitigates difficulties in hostile environments. On the single-cell dataset CIRRL is the clear leader across all three summary statistics, showing both higher average accuracy and far greater robustness than competing approaches. ERM's combination of a small ICU median with a very large mean and worst case reveals a skewed environment distribution. By contrast, CIRRL (and to a lesser extent DRIG) trade a small amount of the performance to deliver more dependable behavior overall. This is a desirable property for biomedical applications where worst-case safety is important. Notably, although simple linear baselines can perform well in biomedical tasks (Londschien et al., 2025; von Kügelgen et al., 2025), CIRRL not only matches or surpasses those baselines but also outperforms every nonlinear alternative we evaluated, offering the best combination of accuracy and robustness. For experiment details, see Appendices E, F.

Table 1: Squared error comparison on real datasets. **Worst** denotes the worst MSE among environments.

| Model | ICU | | | Single-cell data | | |
|---|---|---|---|---|---|---|
| | **Mean** | **Median** | **Worst** | **Mean** | **Median** | **Worst** |
| CIRRL | **0.413** | 0.240 | **2.970** | **0.332** | **0.179** | **0.606** |
| ERM | 3.550 | **0.079** | 37.47 | 0.402 | 0.226 | 0.772 |
| IRM | 2.634 | 1.640 | 18.23 | 0.405 | 0.216 | 0.728 |
| VREX | 0.928 | 0.427 | 5.454 | 0.380 | 0.183 | 0.912 |
| ARM | 0.601 | 0.360 | 3.673 | 0.404 | 0.215 | 0.798 |
| DRIG (linear) | 0.414 | 0.269 | 2.974 | 0.356 | 0.323 | 0.703 |

## 5 CONCLUSION

In this work, we proposed a robust framework for estimating target variables in high-dimensional settings, designed to handle distribution shifts. We demonstrated its effectiveness on both synthetic and real-world datasets, consistently outperforming traditional methods like ERM, IRM, ARM, V-REx, and DRIG. Our synthetic experiments confirmed the model's resilience to distributional variations, including deviations from ellipticity, suggesting potential for broader applications without strict structural assumptions. Future work could explore more complex latent structures, interactions between observed and unobserved variables, and relax the ellipticity assumption. Extensions to non-independent data or sequential environments also present promising directions.

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

## A  SYMBOLS

Table 2: Table of Notation

| SYMBOL | MEANING |
|---|---|
| $\mathbf{B} \in \mathbb{R}^{(k+1) \times (k+1)}$ | ADJACENCY MATRIX OF THE LATENT CAUSAL GRAPH |
| $\mathbf{I}$ | IDENTITY |
| $\phi^*$ | TRUE DECODER FUNCTION |
| $X, X^e$ | OBSERVED COVARIATES, OBSERVED COVARIATES IN ENVIRONMENT $e$, RESPECTIVELY |
| $Z, Z^e$ | LATENT VARIABLES, LATENT VARIABLES IN ENVIRONMENT $e$, RESPECTIVELY |
| $Y, Y^e$ | RESPONSE, RESPONSE IN ENVIRONMENT $e$, RESPECTIVELY |
| $\varepsilon \in \mathbb{R}^{k+1}$ | NOISE VARIABLE |
| $\xi, \tilde{\varepsilon}$ | NOISE VARIABLES FOR $g$, DEC, RESPECTIVELY |
| $\delta^e \in \mathbb{R}^{k+1}$ | INTERVENTION IN ENVIRONMENT $e$ |
| $v \in \mathbb{R}^{k+1}$ | INTERVENTION IN THE TEST ENVIRONMENT |
| $\phi^{\text{OPT}}, \widehat{\phi}$ | OPTIMIZED ENCODER FUNCTION |
| $\text{DEC}^{\text{OPT}}$ | OPTIMIZED DECODER FUNCTION |
| $g^{\text{OPT}}$ | LEARNED AND OPTIMIZED PRIOR ENCODER FUNCTION |
| $\mu_g, \Sigma_g$ | $g$ ESTIMATES OF THE LATENT MEAN AND ROOT COVARIANCE |
| $\widehat{b}$ | ESTIMATED DRIG COEFFICIENT |
| $\gamma$ | PERTURBATION STRENGTH TO ACCOUNT FOR IN DRIG ESTIMATION |
| $C^\gamma$ | UNCERTAINTY SET OF DRIG |
| $\widehat{f}$ | FINAL MODEL |
| $\mathbb{L}^2(\mathcal{A})$ | SPACE OF SQUARE INTEGRABLE FUNCTIONS OVER $\mathcal{A}$ |
| $\eta$ | PERTURBATION STRENGTH IN THE TEST ENVIRONMENT - SIMULATED DATA |

## B  IDENTIFIABILITY

**Definition 1.** *(Weakly injective) Let $f : \mathbb{R}^k \to \mathbb{R}^d$ and $k \le d$ . A function $f$ is said to be weakly injective, if*

- *There is a point $x_0 \in \mathbb{R}^d$ and $\delta > 0$ with $|f^{-1}(x)| = 1$ for all $x$ in the image of $f$ intersected with the $\delta$-ball around $x_0$, **and***

- *the set $\{x \in \mathbb{R}^d : |f^{-1}(x)| = \infty\} \subseteq f(\mathbb{R}^k)$ has Lebesgue measure zero.*

Assuming $P$ follows a Gaussian mixture, together, $E$ with $Z$ encodes the latent structure

$$E = e \sim P_\lambda(E = e)$$

$$Z|E = e \sim \mathcal{N}(\mu_e, \Sigma_e)$$

$$E \to Z \to X$$

where $P_\lambda$ denotes the marginal distribution of $E$ that depends on $\lambda_j$, and $\sum_{j=1}^J \lambda_j = 1$ denote the positive weights of the components in the Gaussian mixture.

For multivariate but discrete $E$, one is required to reconstruct everything just from the distribution of $X$. Under stronger conditions, it is also possible to identify $P(E)$.

**Lemma 4.** *(Kivva et al., 2022) Let the GMM model as defined above hold and let $f$ be piecewise affine. If $f$ is weakly injective, $P(Z)$ is identifiable from $P(X)$ up to an affine transform.*

Ahuja et al. (2023) assume that the interior of the support of $z$ is nonempty subset of $\mathbb{R}^k$, and that the decoder function $f$ is an injective polynomial of finite degree. Also, assuming that the interior of the encoder image is a nonempty subset of $\mathbb{R}^k$

**Lemma 5.** *If the autoencoder solves the reconstruction identity $dec \circ enc(x) = x$ for all $x$ in its support, under the constraint that the learned decoder is a polynomial of the same degree as the true decoder, then it achieves affine identification, ie recovers an affine transformation of the true latent variables.*

For a deeper overview of similar and stronger results, concerning even more general decoders, we refer the reader to Khemakhem et al. (2020b); Kivva et al. (2022); Ahuja et al. (2023); Buchholz et al. (2023).

### B.1 DISCUSSION OF COMPATIBLE ASSUMPTIONS

Although the proposed objective superficially resembles a VAE loss (reconstruction + latent regularization), however, key differences are: (1) We use the DPA objective (energy matching) rather than a KL divergence, which encourages matching distributions rather than likelihood; (2) The prior is environment-conditioned (through $g(E, \xi)$), so we learn a different Gaussian for each $E$; (3) The goal is not just representation learning but causal robustness. In other words, we could view our model as a conditional (Mixture) VAE on $X|E$, but with an energy-based penalty (Eq.(4)) and with a second robust regression step on top. To clarify these points: the VAE viewpoint is useful intuition but CIRRL is explicitly designed for invariance/robustness rather than generative modeling. We refer to Shen & Meinshausen (2024) for more details about this type of loss function and its relations to VAEs.

Furthermore, it is possible to obtain analogous results by relying on Lemma 5 instead of Lemma 2. In the former case, using a VAE would ensure the reconstruction identity; however, the polynomial condition would remain unclear.

## C  PROOFS

*Proof of Proposition 1.* Similar to the proof of Theorem 2 in Shen et al. (2023), replacing $X$ with $\phi^*(X)$. From the SCM, it is known that

$$\phi^*(X^v) = (\mathbf{I} - \mathbf{B})^{-1}_{1:k,\cdot}(\varepsilon + v) \text{ as well as } Y^v = (\mathbf{I} - \mathbf{B})^{-1}_{k+1,\cdot}(\varepsilon + v)$$

and by centering it is clear that, $\phi_c(X^v) = N(\mathbf{I} - \mathbf{B})^{-1}_{1:k,\cdot}(\varepsilon + v) = N\phi^*(X^v)$ while $Y^v_c = Y^v$.

Let us denote $Y^v - b^\top \phi_c(X^v) = \left( (\mathbf{I} - \mathbf{B})^{-1}_{1:k,\cdot} - b^\top N(\mathbf{I} - \mathbf{B})^{-1}_{k+1,\cdot} \right)(\varepsilon + v) = w^\top(\varepsilon + v)$. Then, rewriting the expectation under $\mathbb{P}_v$ gives

$$\mathbb{E}_v[Y - b^\top \phi_c(X)]^2 = \mathbb{E}[w^\top(\varepsilon + v)]^2 \tag{7}$$

$$= w^\top \mathbb{E}[(\varepsilon + v)(\varepsilon + v)^\top]w \tag{8}$$

$$= \mathbb{E}_0[Y - b^\top \phi_c(X)]^2 + w^\top \mathbb{E}[vv^\top]w \tag{9}$$

Taking the supremum over $v \in C^\gamma$ on both sides only influences the second expression of the right hand side

$$\sup_{v \in C^\gamma} w^\top \mathbb{E}_v[vv^\top]w = w^\top \left( S^0 + \gamma \sum_{e \in \mathbb{E}} \omega^e (S^e - S^0 + \mu^e \mu^{e\top}) \right) w \tag{10}$$

$$= \mathbb{E}[w^\top \delta^0]^2 + \gamma \sum_{e \in \mathbb{E}} \omega^e (\mathbb{E}[w^\top \delta^e]^2 - \mathbb{E}[w^\top \delta^0]^2) \tag{11}$$

Noticing that by independence of $\varepsilon, \delta^e$ and the fact that $\varepsilon$ is centered $\mathbb{E}_0[Y - b^\top \phi_c(X)]^2 + \mathbb{E}[w^\top \delta^0]^2 = \mathbb{E}[Y^0 - b^\top \phi_c(X^0)]^2$ holds, as well as

$$\mathbb{E}[w^\top \delta^e]^2 - \mathbb{E}[w^\top \delta^0]^2 = \mathbb{E}[w^\top \delta^e]^2 + \mathbb{E}[\varepsilon^\top \delta^e]^2 - \mathbb{E}[\varepsilon^\top \delta^e]^2 - \mathbb{E}[w^\top \delta^0]^2 \tag{12}$$

$$= \mathbb{E}[Y^e - b^\top \phi_c(X^e)]^2 - \mathbb{E}[Y^0 - b^\top \phi_c(X^0)]^2 \tag{13}$$

for all environments $e \in \mathbb{E}$, finishes the proof. $\qquad\square$

The following technical lemma studies a helpful expression combining affine transforms of a random variable and a conditional expectation in case of normal distribution.

**Definition 2.** (spherically symmetric distribution, elliptical symmetric distribution)

- *A random vector $Y \in \mathbb{R}^d$ follows a spherically symmetric distribution, if there exists a scalar function $\psi$, such that $\psi(u^\top u) = \mathbb{E}[e^{iu^\top Y}]$. In this case, we denote $Y \sim S_d(\psi)$.*

- *A random vector $X \in \mathbb{R}^d$ follows an elliptically symmetric distribution with parameters $\mu, \Sigma, \psi$, if $X = \mu + AY$ in distribution, for $\Sigma = AA^\top$, $A \in \mathbb{R}^{d \times k}$, $rk(A) = k$ and $Y$ following a spherical distribution with a scalar function $\psi$. In this case, we denote $X \sim E_d(\mu, \Sigma, \psi)$.*

Further examples of elliptical distributions include symmetric multivariate Laplace, Kotz, and logistic distribution. For a better overview of their properties, we refer to Fang (1990).

**Lemma 6.** *Let $X \in \mathbb{R}^d$, and $X \sim E_d(\mu, \Sigma, \psi)$. Then, for a full-rank matrix $M \in \mathbb{R}^{k \times d}$ with $k \leq d$, the conditional expectation $\mathbb{E}[X|MX]$ is affine in $MX$, i.e. there is a matrix $A \in \mathbb{R}^{d \times k}$ and a vector $c \in \mathbb{R}^d$ such that*

$$\mathbb{E}[X|MX] = AMX + c.$$

*Proof of Lemma 6.* Since $X \sim E_d(\mu, \Sigma, \psi)$, it is clear that $MX \sim E_k(M\mu, M\Sigma M^\top, \psi)$, and also

$$\begin{pmatrix} X \\ MX \end{pmatrix} \sim E_{d+k} \left( \begin{pmatrix} \mu \\ M\mu \end{pmatrix}, \begin{pmatrix} \Sigma & \Sigma M^\top \\ M\Sigma & M\Sigma M^\top \end{pmatrix}, \psi \right). \tag{14}$$

Furthermore, similar to the Gaussian case, the conditional $X|MX$ also follows an elliptically symmetric distribution and its expectation is known (Fang, 1990),

$$\mathbb{E}[X|MX] = \mu + \Sigma M^\top (M\Sigma M^\top)^{-1} M(X - \mu) \tag{15}$$

$$= \Sigma M^\top (M\Sigma M^\top)^{-1} MX + (\mathbf{I} - \Sigma M^\top (M\Sigma M^\top)^{-1} M)\mu, \tag{16}$$

which is an affine function of $MX$. $\qquad\square$

**Corollary 7.** *In case $X$ is a centered random variable as above, then $\mathbb{E}[X|MX]$ is a linear function of $MX$.*

*Remark 2.* The matrix $\Sigma M^\top (M\Sigma M^\top)^{-1} M$ is an (oblique) projection onto the row space of $M$, transformed by $\Sigma$.

*Proof of Theorem 3.*

$$\sup_{v \in C^\gamma} \mathbb{E}_v[Y - \widehat{b}^\top \widehat{\phi}_c(X)]^2 = \min_{b \in \mathbb{R}^k} \sup_{v \in C^\gamma} \mathbb{E}_v[Y - b^\top N\phi^*(X)]^2 \tag{17}$$

$$= \min_{b \in \mathbb{R}^k} \sup_{v \in C^\gamma} \mathbb{E}_v[Y - b^\top \phi^*(X)]^2 \tag{18}$$

Now consider the minimizer of $\mathbb{E}_v[Y - f(X)]^2$, namely $\mathbb{E}_v[Y|X]$. Since $X, Y$ are $d$-separated by $\phi^*(X)$ in the SCM, we see how $\mathbb{E}_v[Y|X] = \mathbb{E}_v[Y|X, \phi^*(X)] = \mathbb{E}_v[Y|\phi^*(X)]$ and moving forward, it becomes clear that

$$\mathbb{E}_v[Y|\phi^*(X)] = (\mathbf{I} - \mathbf{B})^{-1}_{k+1,\cdot} \mathbb{E}\left[\varepsilon + v \,\middle|\, (\mathbf{I} - \mathbf{B})^{-1}_{1:k,\cdot}(\varepsilon + v)\right] \tag{19}$$

$$\text{by Lemma 6} = (\mathbf{I} - \mathbf{B})^{-1}_{k+1,\cdot} AM(\varepsilon + v) + (\mathbf{I} - \mathbf{B})^{-1}_{k+1,\cdot}(\mathbf{I} - AM)\mu_v, \tag{20}$$

for $A = \Sigma M^\top (M\Sigma M^\top)^{-1}, M = (\mathbf{I} - \mathbf{B})^{-1}_{1:k,\cdot}, \Sigma = \Sigma_\varepsilon + \Sigma_v$. $AM$ is a projection matrix projecting onto the row-space of $M$, transformed by $\Sigma$. In case $\mu_v$ can be represented as $\Sigma M^\top \alpha$ for some $\alpha \in \mathbb{R}^k$, then it is clear that $(\mathbf{I} - AM)\mu_v = 0$ is always the case. Hence, $\mathbb{E}_v[Y|X]$ is a linear function of $\phi^*(X)$ as as whole, it is true that for some $a \in \mathbb{R}^k$

$$\min_{f \in \mathbb{L}^2(\mathcal{X})} \mathbb{E}_v[Y - f(X)]^2 = \mathbb{E}_v\left[Y - \mathbb{E}_v[Y|X]\right]^2 \tag{21}$$

$$= \mathbb{E}_v[Y - a^\top \phi^*(X)]^2 \tag{22}$$

$$\geq \min_{b \in \mathbb{R}^k} \mathbb{E}_v[Y - b^\top \phi^*(X)]^2 \tag{23}$$

The reverse inequality between the two is always true. Since for any $v \in C^\gamma$

$$\min_{f \in \mathbb{L}^2(\mathcal{X})} \mathbb{E}_v[Y - f(X)]^2 = \min_{b \in \mathbb{R}^k} \mathbb{E}_v[Y - b^\top \phi^*(X)]^2,$$

it also holds

$$\min_{f \in \mathbb{L}^2(\mathcal{X})} \sup_{v \in C^\gamma} \mathbb{E}_v[Y - f(X)]^2 = \min_{b \in \mathbb{R}^k} \sup_{v \in C^\gamma} \mathbb{E}_v[Y - b^\top \phi^*(X)]^2.$$

$\square$

## C.1 RELATIONSHIP TO PRIOR WORK

We emphasize that our combination of DPA (or another method like VAE) and DRIG is non-trivial and yields a novel theoretical guarantee in the nonlinear setting. To our knowledge, this is the first causality-inspired DRO method that provides a finite-radius robustness certificate when the covariate-to-response relationship is nonlinear. Theorem 3 in our paper is analogous in spirit to the main result of Shen et al. (2023) but is strictly stronger: it establishes optimality of our predictor over *all* measurable functions (not just linear ones) following an SCM (equations 2, 3). The proof relies on the identifiability of latent representations and the linear latent-to-$Y$ model, which were not handled in prior work.

Our objective superficially resembles a VAE loss (reconstruction + latent regularization). However, key differences are: (1) We use the DPA objective (energy matching) rather than a KL divergence, which encourages matching distributions rather than likelihood; (2) The prior is environment-conditioned (through $g(E, \xi)$), so we learn a different Gaussian for each $E$; (3) The goal is not just representation learning but causal robustness. In other words, we could view our model as a conditional (Mixture) VAE on $X|E$, but with an energy-based penalty and with a second robust regression step on top. The VAE viewpoint is useful intuition but our method is explicitly designed for invariance/robustness rather than generative modeling. We refer to Shen & Meinshausen (2024) for more details about this type of loss function and its relations to VAEs.

Equation 5 looks similar to an IRM/REx-style objective. Indeed, one can think of it as a Lagrangian for a DRO problem: the first term is the (squared-error) loss in the reference environment, and the weighted sum of differences penalizes deviations of other environments from the reference performance. Crucially, however, we derived this form from Proposition 1 (robust optimization) rather than by demanding an invariance

constraint. Concretely, Proposition 1 shows that minimizing our DRO risk leads to exactly the objective in equation 5. In IRM, one would force all environments to have equal risk, while our objective penalizes their squared risk discrepancies.

## C.2 DISCUSSION OF TECHNICAL ASSUMPTIONS

The GMM condition of $(E, Z)$ is a technical assumption to apply the identifiability result lemma 2. In our model, we indeed enforce that is a Gaussian mixture via the prior network, which satisfies lemma 2. In words, lemma 2 says that if $X$ is generated by a piecewise-affine decoder from a latent $Z$ that is Gaussian conditioned on $E$, then one can recover $Z$ up to an affine transformation from $X$. Assumption 1 states that the mean effect of the test intervention $v$ on $Y$ *passes only through* the latent $Z$. Formally $\mathbb{E}[v] = \Sigma[(\mathbf{I} - \mathbf{B})_{1:k,.}^{-1}]^\top \alpha$ means that $\mathbb{E}[v]$ lies in the column space of the latent-to-$Y$ effect rows. Intuitively, it ensures the average shift in can be represented as a linear combination of the latent variables' total effects. In practice, it is used in a proof to reduce an affine expression to a linear one (see, e.g., equation 19 in the proof of theorem 3, Appendix C).

The GMM assumption on $Z|E$ is indeed a modeling choice for two reasons: theory and algorithmic regularization. Theoretically, the GMM is used to guarantee identifiability of up to affine (Lemma 2). In practice, however, performance is not very sensitive to this assumption. For example, in our synthetic experiment (NOTEWHICH), we tested an OOD case where the latent shifts were drawn from a $\chi^2$ distribution (i.e. non-Gaussian). The results were almost identical to the Gaussian case, suggesting CIRRL still works even when the mixture prior is misspecified. The Gaussian prior is mainly a convenient choice (and a limit case of the elliptical assumption in Theorem 3), but the method can handle non-Gaussian data as well. We also note that if $Z|E$ truly followed some other known parametric form, one could adapt our prior network accordingly; but the GMM is a flexible "default". Furthermore, the Gaussian mixture assumption originating from results (Kivva et al., 2022), where they guarantee identifiability in case one learns the distribution. There are also other avenues, using for example VAE would produce full reconstruction at optimal parameters, and there are other identifiability results (Khemakhem et al., 2020a; Ahuja et al., 2023) one could use instead of GMMs.

Lemma 2 and Assumption 1 serve distinct but complementary purposes in our two-stage framework. Lemma 2 is concerned exclusively with representation learning: under the Gaussian-mixture prior and a piecewise-affine decoder, it guarantees that the encoder recovers the true latent variables up to an unknown affine transformation, providing a "consistent" embedding space in which linear methods remain valid. Assumption 1, on the other hand, is a statement about the test-time perturbation in equation 3: it requires that the average intervention vector lie in the subspace defined by the latent-to-output effect, which in turn ensures that the population-optimal predictor $\mathbb{E}[Y|X]$ is a linear function of the latents. They address entirely different aspects, Lemma 2 ensures we can safely apply linear techniques to the learned representation, while assumption 1 ensures that the causal shifts themselves remain linear in that space. They together justify our use of a causality-inspired robust linear estimator on $\widehat{Z}$. In other words, once one has a reliable, affine-equivalent embedding (by Lemma 2), one can invoke results from distributional robustness for linear SCMs (via Assumption 1) to obtain the finite-radius robustness guarantee of the method.

## D HYPERPARAMETERS

The architecture of the baseline (IRM, ARM, VREx, ERM) networks, including depth, width, and hyperparameters such as batch normalization, learning rate, dropout, and the number of training epochs, was designed to match the corresponding networks in DPA. During training, the models were fed batches of $(X, Y)$ pairs, and the network predicted $Y$ directly from $X$. Hyperparameters that have not been changing: width was generally taken to be 400 units in all networks as well as a batch norm. In the DPA all networks had the same depth of two in experiments with all datasets and four for studies on single-cell data. All experiments

used Adam for optimization, with a learning rate $10^{-4}$ and $\alpha = 10^{-1}$. The latent dimension was chosen considering a point of type *elbow* (see Remark 4), after which there was no significant performance gain in terms of the objective function $L$. Most experiments do not require significant memory to run as the models are essentially common neural networks and using a GPU usually take between 7 and 12 minutes for 1000 epochs of training.

In all experiments, VREx and IRM shown for best $\beta$ and $\lambda$, respectively, $\in \{\frac{1}{100}, \frac{1}{10}, 1, 10, 100\}$, ARM for best $T \in \{\frac{1}{10}, \frac{1}{2}, 1, 2, 5\}$, CIRRL and DRIG for $\gamma = 15$ in real data experiments and $\gamma = 3$ in artificial ones. For comments on the choice of $\gamma$ and other hyperparameters, see Appendix D.

*Remark* 3 (Hyperparameter $\gamma$). Across our experiments a recurring pattern is CIRRL's low sensitivity to the robustness radius $\gamma$ once $\gamma$ is sufficiently large. Only the small values of $\gamma$ (for example $0, 1, 2$) produce noticeable changes in performance; these cases can be viewed as borderline model misspecifications. Concretely, $\gamma = 0$ reduces the estimator to one that effectively ignores all environments except the reference environment, while $\gamma = 1$ treats interventional and reference environments with equal weight. Because intervened environments often contain far fewer samples than the reference environment (for example in single-cell), small choices of $\gamma$ fail to reflect the intended robustness objective and the nature of the data: they underweight the contribution of interventions and therefore do not exploit the invariances or stable features the method is designed to recover.

*Remark* 4 (Latent dimension $k$). In our experiments we chose the latent dimension $k$ using the "elbow" method on the training loss curve (Figure 5): performance jumped from $k = 1$ to $k = 2$ and then plateaued. In fact, as shown in the synthetic experiment, adding more than two latent factors did not improve performance, so we picked the smallest at which the curve flattened. In practice one trains the DPA using the largest considered latent dimension and then selects the the first dimensions with the most variance. A visual-elbow rule is common and computationally easy. Importantly, Theorem 3 does not require knowing $k$ exactly. Our analysis (Proposition 1 and Theorem 3) holds for any affine transformation of the true latent vector, so if is overestimated, the extra dimensions can simply be ignored by the linear predictor (their weights will be zero or irrelevant).

# E  SYNTHETICALLY GENERATED DATA

It is often helpful and insightful to test new methods and theories in controlled settings, for example, in simulated or lab environments, to ensure that everything works as expected. Let $X \in \mathbb{R}^d$, let $Z \in \mathbb{R}^k$, and let $Y$ be a real variable. As before, it is implicitly assumed that $k \leq d$, as well as $d \geq 2$. Let $\phi^* : \mathbb{R}^d \to \mathbb{R}^k$ denote the true encoder function. The aim is to generate data from environments according to the causal graph similar to the one in subsection 3.1. The true encoder function in synthetic experiments is chosen as a polynomial, parametrized through a random coefficient matrix which adheres to the constraints described in Ahuja et al. (2023) section 4.

That is, $X$ admits a lower dimensional representation $Z = \phi^*(X)$, which is a nonlinear function of $X$, while $Y$ itself is a linear function of $Z$. For an environment $e \in \mathcal{E}$, the corresponding modified version of the structural equations as noted in equation 2 is

$$\begin{pmatrix} \phi^*(X^e) \\ Y^e \end{pmatrix} = \mathbf{B} \begin{pmatrix} \phi^*(X^e) \\ Y^e \end{pmatrix} + \varepsilon + \delta^e \tag{24}$$

where $e \in \mathcal{E}$ and $e = 0$ denotes the observational environment and $\delta^e$ denotes the intervention which is independent of $\varepsilon$. Analogously, the test setting, for $v$ also independent of $\varepsilon$, as in equation 3.

$$\begin{pmatrix} \phi^*(X^v) \\ Y^v \end{pmatrix} = \mathbf{B} \begin{pmatrix} \phi^*(X^v) \\ Y^v \end{pmatrix} + \varepsilon + v \tag{25}$$

To begin, a random directed graph $G$ is generated, with edge probabilities of $\frac{1}{2}$. To enforce the acyclic condition, the edges of the random graph $G$ are filtered to eliminate cycles. In particular, the new graph will

only include edges where the source node has a *higher* index than the target node. The remaining edges are then assigned a weight drawn from the normal distribution. Let $\mathbf{B}$ denote the adjacency matrix of this directed acyclic graph (DAG) as above, and let $\mathbf{C}$ denote its inverse $(\mathbf{I} - \mathbf{B})^{-1}$. Let $b$ denote the first $k$ components of the $(k+1)$-st row, $\mathbf{B}_{k+1,1:k}$.

Next, random positive definite matrices with norm 1 are generated to serve as the covariance matrices $\Sigma_e$ of the interventions $\delta^e$ in each environment, as well as normalised random vectors $\mu_e$ to be used as their means. The second moment of the test intervention $v$ is then derived from the covariance matrices and means of the individual interventions

$$\Xi_\eta = \frac{\eta}{|\mathcal{E}|} \sum_{e \in \mathcal{E}} \left( \Sigma_e + \mu_e \mu_e^\top \right)$$

and is controlled by the parameter $\eta$, which is given as input. The covariance matrix of $\varepsilon$ is generated similarly to those of $\delta^e$. The role of $\eta$ is to control the strength of perturbation within the test set.

Furthermore, the means and covariance matrices are used to sample $\varepsilon$ as well as $\delta^e$ from the normal distribution, for each environment $e$. The data points are subsequently generated as

$$\begin{pmatrix} Z^e \\ Y^e \end{pmatrix} = \mathbf{C}(\varepsilon + \delta^e). \tag{26}$$

Lastly, $X^e$ is obtained by plugging $Z^e$ into the specified latent function. The latent function can be taken as a polynomial of given degree, with some constraints on the dimensionalities of $X$ or $Z$ as mentioned in Ahuja et al. Ahuja et al. (2023).

Alternatively, the latent function can also be chosen as an initialised ReLU network of given width and depth to cosplay as a nonlinear function. Training data points are collected, as well as their labeled environments encoded in $E$, and are divided in batches to be used in training. Concerning the data for the test environment, $\varepsilon$ is sampled analogously as in the training setting, but $v$ is sampled as a Gaussian with mean $\mu_v$ and covariance $\Xi_\eta - \mu_v \mu_v^\top$ where $\mu_v$ can be given as input or it defaults to $\mu_v = \frac{\eta}{|\mathcal{E}|} \sum_{e \in \mathcal{E}} \mu_e$. Also similarly to the train setting, the data points are subsequently generated as

$$\begin{pmatrix} Z^v \\ Y^v \end{pmatrix} = \mathbf{C}(\varepsilon + v). \tag{27}$$

and $X^v$ is obtained after plugging $Z^v$ into the same latent function. Furthermore, in case of student's $t$-distribution, the vector $\varepsilon + v$ is derived with one additional step before multiplication by $\mathbf{C}$. Taking $\varepsilon + v$ generated as above, denote it by $\zeta$. Since $\zeta$ is Gaussian, the vector

$$\zeta' := \zeta_c \cdot \sqrt{\frac{\nu}{u}} + \text{mean}_\zeta, \quad u \sim \chi_\nu^2,$$

where $\zeta_c$ is simply $\zeta$ centered, follows a multivariate student's $t$-distribution with $\nu$ degrees of freedom. By abuse of notation, denote $\zeta'$ by $\varepsilon + v$ and plug it into equation 27.

Additionally, since the same number of observations is drawn from each environment, the weights $\omega^e$ simplify to the uniform weights in this context. There is also an option to consider the setting where one excludes any intervention on the target variable $Y$, this would mean that the means and covariance matrices of $\delta^e, v$ are generated with a last entry, respectively, row being zero, and the matrices are in that case only positive semidefinite.

The submitted code also contains an oracle estimator, namely the MSE of DRIG applied on the true latent variables, which is considered to be the theoretically optimal achievable DRIGs MSE in case of perfect reconstruction. Furthermore, the experimental data was drawn using perturbation strength $\eta = 10$.

### E.1 SYNTHETIC EXPERIMENT DETAILS

In simulated experiments, the data was generated according to the SCM introduced in Subsection 3.1, and the technical details are described in Appendix E. We sample one observational and four interventional environments with 2000 samples, each, from a two-dimensional latent variable model and embed them in 10 dimensional space using a polynomial decoder function. Out-of-distribution (OOD), or the test set, is generated with a fixed perturbation strength (Appendix E), with latents drawn from a Gaussian and $\chi^2$ for the well-specified and misspecified case, respectively.

### E.2 SINGLE-CELL DATASET

We evaluated the method using a large single-cell RNA sequencing dataset Replogle et al. (2022). This dataset involves genome-wide CRISPR-based perturbations performed on millions of human cells to systematically target expressed genes. For our analysis, we focus specifically on the subset of data derived from RPE1 cells, which emphasize genes likely to play critical roles and exhibit responsiveness to interventions. Following the preprocessing steps established by Chevalley et al. (2022), we selected the 10 genes with the highest expression levels to serve as observed variables. Among these, one gene is treated as the response variable, while the other nine are treated as predictors, based on the reasoning provided by Shen et al. (2023). Our training set consists of 11,485 reference samples, supplemented by data from 10 distinct interventional environments, each corresponding to a targeted perturbation of one of the observed genes. The number of samples per intervention varies between 100 and 500. Furthermore, the dataset includes numerous additional environments, where interventions are applied to unobserved genes outside the set of 10 selected genes. These unseen environments, which differ from those in the training set, are used as test scenarios to evaluate the robustness of prediction models across diverse distributions.

### E.3 INTENSIVE CARE UNIT DATASET

We further evaluate our model using two large intensive care unit (ICU) electronic health record repositories: MIMIC-III (Johnson et al., 2019; Bennett et al., 2023; Londschien et al., 2025), which contains de-identified ICU admissions from Beth Israel Deaconess Medical Center, and the multi-site eICU database (Johnson et al., 2021), which aggregates ICU records from U.S. hospitals other than the MIMIC-III site. The prediction target is each patient's mean heart rate measured during the 48–72 hour window after ICU admission. Predictor variables include patient demographics together with a curated set of clinical and laboratory measurements. After preprocessing, the analysis uses 31 covariates, with 784 eICU records partitioned into four region-based training environments and 67 MIMIC-III records reserved for testing. Because the observational (ignorability) assumption does not hold in this setting, our focus is on learning models that generalize across environments.

We select the 17 covariates which show less than 10% of missing values: blood urea nitrogen (bun), calcium (ca), chloride (cl), creatinine (crea), glucose (glu), hemoglobin (hgb), heart rate (hr), potassium (k), mean arterial pressure (map), sodium (na), oxygen saturation (o2sat), respiratory rate (resp), white blood cell count (wbc), age, sex, height, and weight. Among these, we impute the missing values with zero, and add an indicator variable to denote the imputed rows, which produces 14 new variables. The training environments are taken as regions in eICU. All 31 variables are used to predict the outcome.

## F FURTHER PLOTS OF EXPERIMENTS

### F.1 WHY IRM UNDERPERFORMS

In our experiments, we observed that IRM gave little or no improvement over ERM. We believe this is because IRM enforces "exact invariance" which may not align with the causal structure of our latent model:

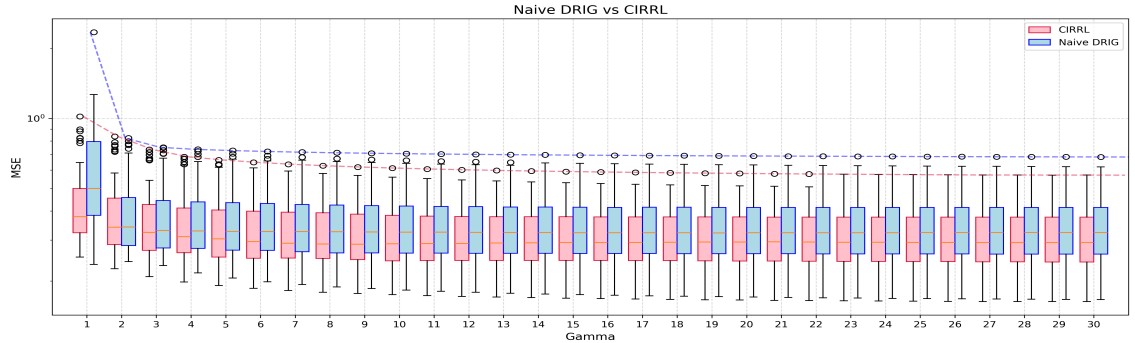

Figure 2: Single-cell dataset. Figure presents boxplots of the MSE across environments. In each group of four boxes—each corresponding to a specific value of $\gamma$ (as indicated on the $x$-axis).Note that both IRM and ERM do not depend on $\gamma$, which is why we exclude them from this figure. The dotted lines overlaid on each group indicate the worst performing environment, marking the maximum MSE error (or worst quantile) observed.

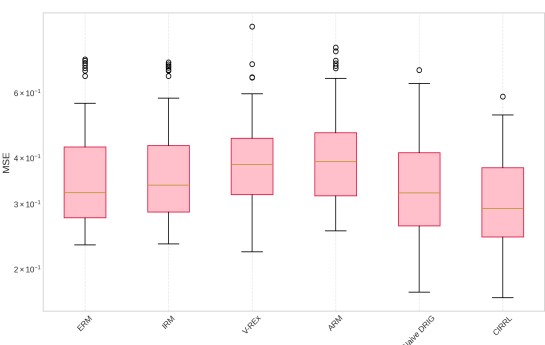

Figure 3: Boxplots for environment MSEs on single-cell test data for all models considered on a log scale. Naive DRIG, CIRRL are evaluated for $\gamma = 15$.

IRM can fail when there is no strictly invariant linear predictor in the chosen representation space. By contrast, our DRO formulation (CIR) finds the predictor that minimizes worst-case risk under bounded latent shifts. IRM's invariance criterion can lead to trivial solutions in this setting, whereas our method seeks a robust-but-not-overly-conservative solution.

In principle, our synthetic SCM admits a single invariant linear predictor once the true latent factors are recovered. However, IRM must learn both a representation and a predictor simultaneously, without any mechanism to guarantee that aligns with the true encoder . In practice, IRM's pressure to enforce exact invariance across only a few finite "environment" shifts causes it to collapse to trivial or near-ERM solutions: the learned typically entangles causal factors so that no single can achieve optimality in every environment.

More fundamentally, IRM's invariance criterion effectively demands robustness under all possible shifts, which is an unbounded requirement, whereas our finite radius (compare figure 1) perspective recognizes that training environments arise from a specific, limited set of perturbations. When only moderate, discrete shifts are observed, insisting on perfect invariance can be too stringent: it forces the predictor to guard against hypothetical changes far beyond those seen in data, often at the cost of worse performance on realistic

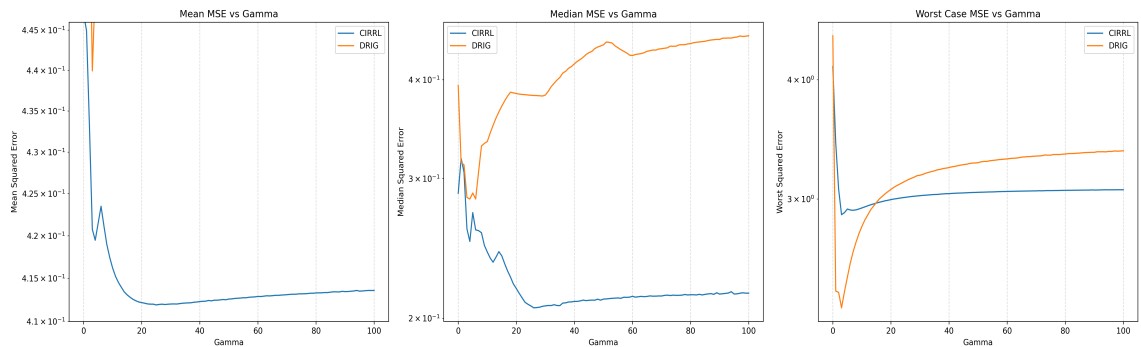

Figure 4: ICU dataset DRIG, CIRRL dependence on $\gamma$.

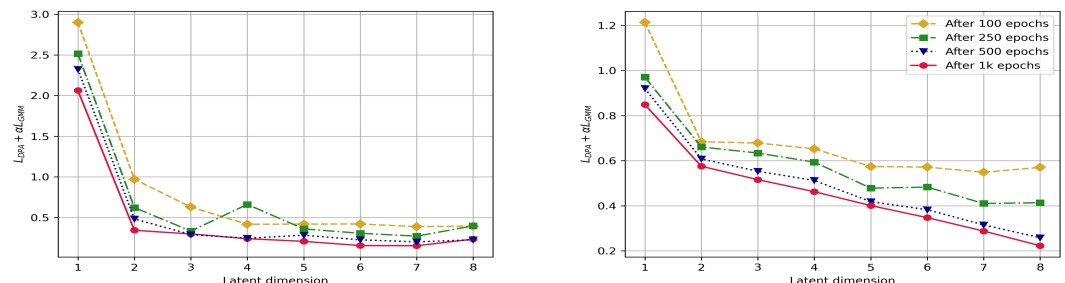

Figure 5: Simulated (left), and single-cell (right) dataset - values of optimized loss function $L$ (equation 4). In case there is a *clear* elbow point, as in the left figure, one should choose it as the latent dimension for the model. However, if the elbow point is less pronounced as it is the case in the second figure, we recommend choosing a value slightly to its right, in this example three or four.

shifts. Finite robustness, by contrast, tailors the worst-case optimization to the actual perturbation magnitude. Empirically, we observe IRM offering little to no improvement over ERM on our synthetic data, even though an invariant predictor exists in principle, because its invariance penalty is too brittle for finite, discrete-sample shifts. This accords with Rosenfeld et al. (2020), who demonstrate that IRM can "fail catastrophically" and often yields no benefit over ERM when only moderate, finite shifts occur.

## F.2 ERROR BARS OF EXPERIMENTS AND ABLATION STUDIES

The GMM regularization is partly motivated by the results of Kivva et al. (2022), who rely on the framework to generating the latents from a GMM. It can also be interpreted as a way to ensure a common invariant structure among the environments. For an ablation study of $\alpha$, consider the figure below.

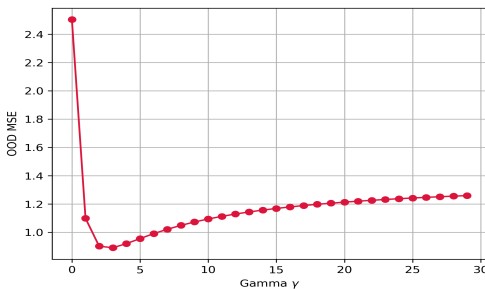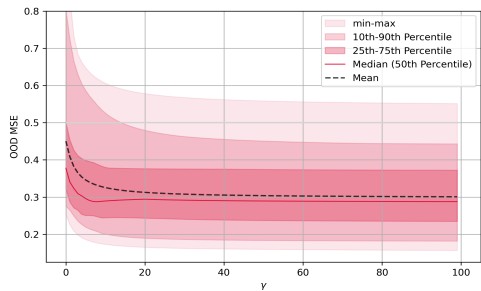

Figure 6: Synthetically generated dataset of latent dimension two (left) & single-cell dataset (right) - the panels illustrate the evolution of OOD MSE error of the proposed model in terms of chosen robustness radius $\gamma$. Notably, in the first case ,the *finite* nature of the perturbation is clear, as the performance degrades for overly conservative values of $\gamma$. This occurrence is less clear in the second case, but still visible in the median. For the right panel, from top to bottom in shades of red: maximum, 90th, 75th quantile, median, 25th, 10th quantile, and minimum of the MSE across test environments. The black dashed line represents the mean.

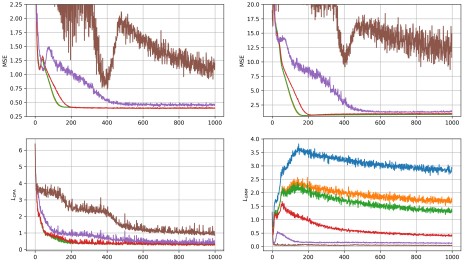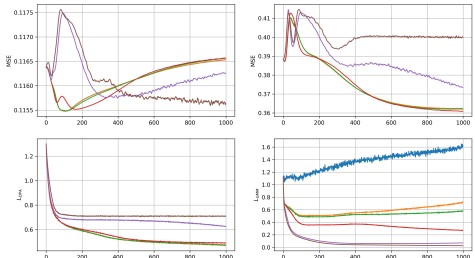

Figure 7: Simulated (left), single-cell (right), learning curves over 1000 epochs for different values of $\alpha \in \{0, \frac{1}{1000}, \frac{1}{100}, \frac{1}{10}, 1, 10\}$ colored blue, orange, green, red, purple, brown, respectively. Considering only the lower pairs of plots, it is evident that $\alpha = \frac{1}{10}$ (red) achieves the best trade-off among the selected values for the optimized loss $L$. The upper row depicts performance in terms of training and test MSE, respectively.

# G  LLM USAGE

We used ChatGPT 4 and Claude Sonet 4 for grammar and phrasing improvements, improving code for experiments, and summarizing certain sections into more compact paragraphs, all of which have been fully revised by the authors. No LLM was used for data analysis, results generation, or core technical contribution.

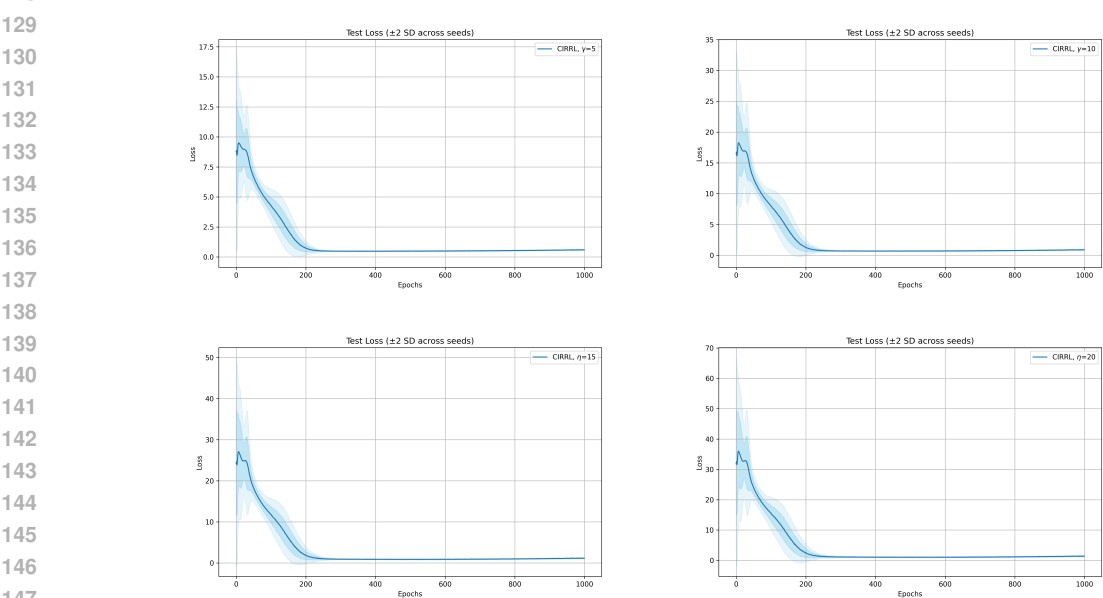

Figure 8: Synthetic data; test loss progression during training with bands denoting one and two standard deviation gaps across 10 seeds for $\gamma = 3$ and $\eta$ values of 5,10,15,20.

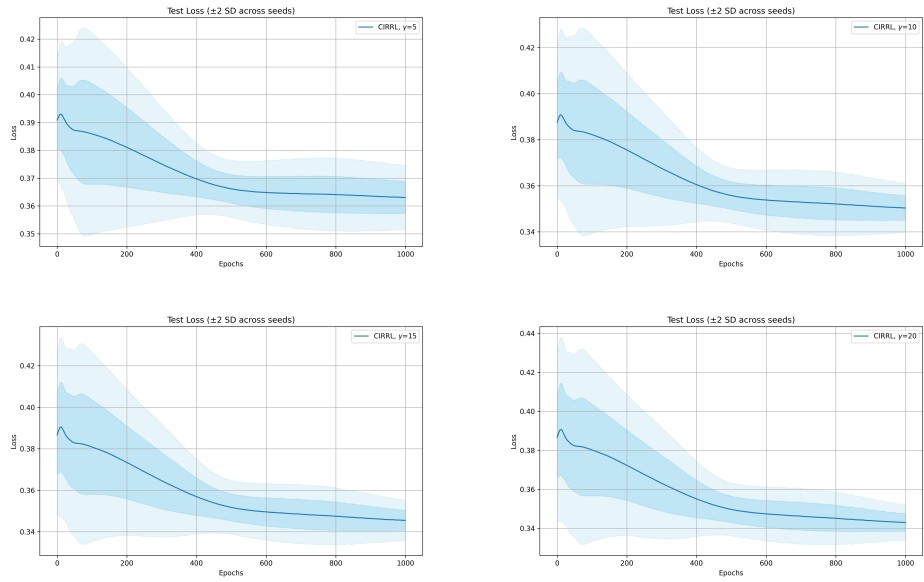

Figure 9: Single-cell; test loss progression during training with bands denoting one and two standard deviation gaps across 10 seeds for $\gamma$ values of 5,10,15,20.

