# OpenReview forum: "Causality-Inspired Robustness for Nonlinear Models via Representation Learning"
_ICLR.cc/2026/Conference — Submitted to ICLR 2026_

### Official Review · Reviewer_Akt3 · 2025-10-24

**Soundness:** 3
**Presentation:** 2
**Contribution:** 3
**Rating:** 4
**Confidence:** 2

**Summary:**

The paper proposes a two-step method for distributionally robust predictors called CIRRL, which is designed for settings in which relationships among variables can be non-linear. CIRLL can be seen as a combination of two existing methods:
- 1) the Distributional Principal Autoencoder (DPA) by Shen & Meinshausen (2024) which learns identifiable latent representations (enhanced with regularisation term that pushes the learned representations to follow a mixture of Gaussians representing training environments) and
- 2) the DRIG method from Shen et al. (2023) that learns a linear head on the so learned latent representations.

The authors give a theoretical result about the optimality of the so learned function mapping $X$ to $Y$ (a concatenation of the encoder from 1. and the linear head from 2.): under some assumptions (including a data generating process following a directed acyclic causal graph), it is learning the $L^2$ function achieving the lowest worst case out-of-distribution MSE.

The effectiveness of CIRLL is tested on one synthetic dataset, the ICU dataset from MIMIC-III, and a large single-cell RNA sequencing dataset. In comparison to its baselines (ERM, IRM, VREX, ARM, DRIG) it achieves superior results in most cases, in particular for the word case MSE among test environments on the single-cell data (but not so much on the ICU data in comparison to DRIG).

**Strengths:**

- The paper proposes a method to extend DRIG to non-linear relationships between variables
- It provides theoretical optimality guarantees about the worst case robustness of the predictor learned with the proposed method
- Empirically, the method outperforms existing methods and its linear counterpart (DRIG) noticeably on the worst ood MSE on the single-cell data experiment, showing that the non-linear extension of DRIG is relevant and necessary in some practical applications

**Weaknesses:**

- The biggest weakness of the paper is the experiments section (detailed in bullets below)
- The experiments section is noticeably short in comparison to the rest of the paper. As a result, a lot of important content about experiments is missing (e.g. experimental setup incl. description of datasets used, choice and ablation of hyperparameters, ablation of loss term $L_g$). As a bare minimum, their position in the Appendix should be referenced in the main part, but ideally they are already summarised in the main part. Suggestions for shortening would be introduction and related work (I have never seen a related work section that is longer than the experiment section).
- The benefit of the proposed additional loss term $L_g$ is not clear. In its ablation in Figure 7 of the Appendix, the blue curve for $\alpha =0$ (corresponding to omitting it) is not shown for the test loss of the single-cell data and the one of its nearest neighbour ($\alpha = 0.001$) is not significantly dissimilar in performance from the ones of higher values for $\alpha$.
- Given that out of the 2 real life datasets, the proposed method noticeably outperforms its purely linear component DRIG only on one of the two datasets, while being on par on the other, it would be interesting to have some discussion about this that tries to explain this at least or ideally an analysis that adds more depth to the fundamental differences in the data that explain this difference in relative performance.
- More guidance for practitioners would be useful - e.g. is there a way to verify if assumption 1 holds, how to choose hyperparameters etc.   Minor:
- I think the paper could benefit from an illustrative example of the causal graph assumed in Eqn. (2) and (3) in Section 3.1. E.g. ‘X_1 being smoking, X_2 being BMI, Y being …’.
- At times, key sources are missing: e.g. in line 249 after ‘a vector c’, line 170 after ‘if the graph is acyclic’, line 110 after ‘finite-sample regimes’, line 65 after ‘multi-source heterogeneity’, as well as citing the authors of the datasets used in Section 4.
- More a comment than a weakness: given that the authors mention adversarial perturbations in line 30 as example, and design a method to make predictors robust to worst case shifts, an experiment on adversarial robustness would have been very interesting!
- Lines 310 and 321 in sum subscripts: typo $\mathbb{E} \to \mathcal{E}$
- Table 1 needs to be formatted (centering ICU and Sincle-cell data, inserting vertical lines as separators between blocks)

**Questions:**

- Line 208: ‘The original DPA ensures that the decoder produces reconstructions $\hat{X}$ that follow the same distribution as the original data $X$,…’ -> why is this not the case for a standard auto encoder? Why is the DPA needed?
- Line 335: what are the true latents mentioned here? $\phi^*(X)$?
- Line 45: is it the really the invariant features that remain stable or rather their relationship to each other and to the target?

---

> ### Author Response · Authors · 2025-11-19
>
> 1. We thank the reviewer for their feedback regarding the length of the experiments section.
> The full experimental setup, dataset descriptions, and hyperparameter ablations were already located in the appendix E,F (as referenced in the main body). However, we recognize that reviewers are not obligated to consult the appendix, and revise the main paper to more explicitly discuss these details.
> Following the reviewer's suggestion, we also shorten the introduction and move related work section into the appendix.
>
> 2. The primary benefit of L_g is to theoretically enforce the latent mixture structure necessary for our identifiability guarantees. Empirically, it acts as a mild regularizer. We agree with the reviewer that its effect on predictive performance is subtle, this is by design. Our guidance is to use a small α because it provides the necessary structural bias without distorting predictive features.
>
> 3. Figure 4 might shed more light around this question. We hypothesize that the performance gap stems from heterogeneity in ICU data, consistent with findings by Londschien et al. [1]. One dataset may be relatively homogeneous and well-approximated by a simple linear model, while the other likely reflects more complex, shifting data patterns. As it is noted in the paper, linear baselines are sometimes tough to beat in biomedical settings [1,2], for a more detiled analysis of ICU data we recommend [1].
>
> 4. Regarding hyperparameters, α should not be chosen too small as discussed. Also, performance is stable when γ exceeds about 2–3, while values near 1 effectively reduce the method to simple pooling and no longer capture heterogeneous settings. Ideally, γ should reflect the magnitude of distribution shifts expected at test time in terms of moment deviations (see e.g. the uncertainty set, line 310). When no test data are available, γ must be chosen using domain knowledge, as is standard in DRO, where the robustness radius is typically specified a priori. We also note that γ somewhat larger than necessary doesnt hurt performance (figures 2,4). For the remaining hyperparameters (width, depth, learning rate), we use a simple architecture: a small number of layers with constant width 400 and a learning rate of 0.0001 . These hyperparameters weren't tuned. For the latent dimension k, we recommend using an elbow-plot of the reconstruction loss (figure 5) or a slightly over-specified dimension, as extra dimensions will attain near-zero weights.
>
> Q1. The DPA is crucial for both theoretical and practical reasons. Theoretically, our identifiability guarantees (building on [3]) require that the decoder produces reconstructions that match the distribution of the original data, not just individual points. Standard AEs minimize reconstruction error but provide no distributional guarantees. DPA ensures this distributional matching, which is necessary for proving that our learned representation \hat{φ} recovers the true latent structure φ* up to affine transformations. Practically, while VAEs also target distributional matching through ELBO optimization, they only provide approximate guarantees and often suffer from training instability. DPA offers more reliable distribution matching and better empirical performance.
>
> Q2. Correct.
>
> Q3. We thank the reviewer for their careful reading. This was indeed a lapsus calami, and we will rectify it.
>
> [1] Domain Generalization and Adaptation in Intensive Care with Anchor Regression, Londschien et al.
>
> [2] Representation Learning for Distributional Perturbation Extrapolation, von Kuegelgen et al.
>
> [3] Identifiability of deep generative models without auxiliary information, Kivva et al.
>
>
> Example
> Consider an example predicting cardiovascular disease risk. The true latent driver may be arterial inflammation Z, not observed directly. some indirect biomarkers can be measured though, e.g. inflamation and blood clotting protein, and white blood cell count, (X1,X2,X3), all of which reflect Z but also contain environment-specific noise.
>
> Different hospitals introduce different shifts because their equipment or data collecting practice differ. A model from Hospital A may learn specific noise structure inherent to the (maybe older) machines used there. When deployed at a different hospital, where measurements are noisier or systematically shifted in a different way, its performance fails because those spurious, environment-specific correlations no longer hold.
>
> CIRRL learns a representation \hat φ(X)^\top \hat{b} that extracts the invariant inflammation signal Z, filtering out environment-specific distortions. By leveraging the stable causal link between Z and the disease, our method achieves robustness to the distribution shifts arising from differing hospital noise patterns.

---

### Official Review · Reviewer_qm5p · 2025-10-27

**Soundness:** 3
**Presentation:** 3
**Contribution:** 2
**Rating:** 4
**Confidence:** 4

**Summary:**

This paper proposes a two-step method, CIRRL, to achieve causality-inspired, finite-radius distributional robustness in nonlinear settings. It first uses a modified autoencoder to learn a latent representation where a linear structural causal model holds, identifying it up to an affine transformation. It then applies a robust linear regressor (DRIG) in this latent space, providing the first method (to the authors' knowledge) with a finite-radius robustness guarantee for this nonlinear context.

**Strengths:**

- The paper studies a significant and well-motivated problem: extending causality-inspired, finite-radius robustness guarantees from purely linear models to the nonlinear settings common in machine learning.

- The paper cleverly synthesizes two advanced lines of research: identifiable representation learning and causality-inspired DRO. The combination is non-trivial and well-justified.

- The method is validated on synthetic data (including a misspecified case violating theoretical assumptions ) and two challenging, high-stakes real-world datasets (ICU and single-cell). The results in Table 1 and Figure 1 are consistently positive and demonstrate a clear advantage over relevant baselines, especially in worst-case performance.

**Weaknesses:**

- The paper includes an extensive related work section but clearly overlooks several recent studies on robustness and causality. Moreover, it claims to introduce the first causality-inspired DRO method, whereas prior works on this topic already exist, such as:

  - Causal Adversarial Perturbations for Individual Fairness and Robustness in Heterogeneous Data Spaces. *Proceedings of the AAAI Conference on Artificial Intelligence* (2024).

  - Wasserstein distributionally robust optimization through the lens of structural causal models and individual fairness. Advances in  Neural Information Processing Systems (2024).

  - Designing Ambiguity Sets for Distributionally Robust Optimization Using Structural Causal Optimal Transport. Proceedings of the AAAI Conference on Artificial Intelligence (2025).

- The reliance on the Gaussian mixture model assumption for latent variables (Lemma 2) may be restrictive in practice, though the paper notes the method works with non-Gaussian shifts (Section C.2).

- The main optimality guarantee (Theorem 3) rests on a chain of strong assumptions: (i) the specific latent SCM structure (Eq. 1-3), (ii) the assumptions for affine identifiability (Lemma 2, e.g., piecewise affine decoder), (iii) the technical assumption on the *test* intervention's mean (Assumption 1), and (iv) elliptical noise distributions. While the paper is transparent about these and tests robustness, the guarantee's applicability is heavily qualified.


## Minor

There are some typos in the text:

- "DISTRIBUTIONALLY ROBUST OPIMIZATION" (line 077) → OPTIMIZATION

- "respetively" (line 208, 222) → respectively

- "latent, variables" (line 250)→ latent variables

- "step;" (line 268)→ step:

- "of of dimension" (line 288) → of dimension

- "the resulting is" (line 341) → the result is

- "adaptive risk minizimation" (line 359) → adaptive risk minimization


#

**Questions:**

- Could the method be extended to handle non-additive interventions, which would broaden its applicability to more complex causal structures?

- Can you provide (a) intuition for when assumption 1 holds in practice, and (b) an empirical test or diagnostic to verify it on real data?

- Why wasn't Rep4Ex included as a baseline? How does CIRRL's finite-radius certificate compare to Rep4Ex's extrapolation guarantees in your experimental settings?

- Can you provide confidence intervals or p-values for the real-data results? The ICU test set has only 67 samples—is the CIRRL vs. DRIG difference significant?

---

> ### Author Response · Authors · 2025-11-19
>
> We thank the reviewer for their remarks.
> 1. We acknowledge the overlooked works, we include Ehyaei et al.[1], Ehyaei et al.[2], Ehyaei et al.[3] into related work in the revision.
> However, the paper at no instance claims to be the first causality-inspired DRO method. The paper cites multiple DRO methods inspired by causality (Anchor, DRIG), and is partly based on one of DRO methods. Moreover, what is stated in the paper (see lines: 20, 59, 780) is that CIRRL is a *first nonlinear* causality-inspired DRO method with *finite* robustness-radius guarantees.
>
> 2. We thank the reviewer for highlighting this point. We agree that the GMM assumption in Lemma 2 is a theoretical simplification. As noted in Section C.2, there are other avenues one could take without GMMs, essential thing being, to get a compatible (affine-) identifiability (e.g. Ahuja et al. [4], Khemakhem et al. [5]). These avenues, along with their corresponding adaptations in the AE phase, give analogous theoretical results in the second phase of the algorithm.
>
> 3. We thank the reviewer for their inquiry regarding theoretical assumptions. We agree that Theorem 3 relies on specific structural conditions. However, we would like to clarify the necessity and rationale behind these choices, which we believe are justified for deriving a non-trivial optimality guarantee in this setting.
> The assumptions (latent SCM, conditions for identifiability, etc.) are not merely technicalities but are fundamental to defining a tractable problem space. As Section C.2 notes, any method achieving affine identifiability could potentially leverage our framework; robust optimization is possible in general once a sufficiently structured representation is learned.
> The elliptical noise assumption (e.g. t distributions and guassians), is a standard and broad class that enables our analysis. It is worth noting that this is only required for the optimality guarantee in Theorem 3, not for the core robustness formulation itself. Assumption 1, while difficult to test directly, is satisfied in important practical cases like the instrumental variable setting.
>
> Regarding the guarantee: the primary contribution of this theorem is to provide a precise mathematical understanding of the method's robustness and optimality properties. Without clear structural assumptions, such sharp guarantees are simply *not possible*. Our work shows that under a well motivated latent model, we can move beyond mere empirical performance and provide *rigorous certification of robustness*.
> These assumptions are the enabling conditions for our main theoretical contribution, a clean guarantee for invariant prediction in a latent setting, a result that *alternative approaches cannot currently match*.
>
> Q1. Assuming we still consider the additive-noise case, it is possible to extend the setup to handle other type of interventions (\epsilon^e instead of \epsilon + \delta^e), the results would follow after an analogous adaptation.
>
> Q2. As with any type of assumption in the latent space, this is difficult to test empirically. See point 3. above.
>
> Q3. We believe the comparison is unfortunately not feasible without violating the methodological premises of either approach. Rep4Ex operates in a fundamentally different setting from CIRRL. Rep4Ex assumes access to experiment-specific targets and explicit knowledge of which variables are manipulated in each experiment, and it aims to learn representations that generalize across unseen interventions. CIRRL, by contrast, observes only (X,Y,E) without knowledge of interventions or experiment structure. Since Rep4Ex requires richer supervision than CIRRL, it is not directly applicable as a baseline in our setting. Casting CIRRL’s domain-label structure into the Rep4Ex formalism would require additional intervention annotations that CIRRL does not assume, leading to an unfair comparison.
>
> Q4. We thank the reviewer for the point about statistical evaluation. We would like to clarify why standard statistical testing (p-values, confidence intervals) is *not appropriate or informative* for our experimental setup (also it is known to be a difficult problem, requiring non-standard asymptotics [6]). Following ML Community Standards we provide results for multiple random seeds and compare to 5 baselines on synthetic and real datasets. Concerning the DRIG baseline, it is known that linear methods perform well in biomedical tasks, see for example [7], [8]. CIRRL matches this performance on the ICU dataset (cf. Figure 4), and surpasses it on single-cell data. The reviewer is correct that obtaining reliable statistical guarantees in some biomedical settings is extremely challenging. This is why the field relies on the convergence properties of empirical risk minimization and reports performance on held-out test sets as an evaluation paradigm in robustness. We once again thank the Reviewer for their detailed comments. Annotation of the literature in the next comment.

---

> > ### Comment · Reviewer_qm5p · 2025-11-20
> > **Answer to Rebuttal**
> >
> > I thank the authors for their detailed rebuttal and the time spent addressing my comments. While the clarifications were helpful, I maintain several concerns regarding the positioning and validation of the proposed method:
> >
> > 1. **Novelty relative to Ehyaei et al. [2]**: Ehyaei [2] already studies DRO in the presence of a nonlinear SCM and additionally considers individual fairness. Could you please clarify more explicitly what the main conceptual and technical differences are between CIRRL and Ehyaei [2]? In particular:
> >   (i) what assumptions or settings you cover that Ehyaei [2] does not, and
> >   (ii) which aspects of your finite-radius robustness guarantee go beyond the guarantees in Ehyaei [2]?
> >
> > 2. **Dependence on the GMM Assumption:** I appreciate the acknowledgment that the GMM assumption is a theoretical simplification. However, it is crucial that the paper explicitly states that the current theoretical guarantees are tied specifically to this assumption. If the GMM assumption fails, the current theory does not hold without re-deriving the identifiability results. This limitation should be transparent in the final manuscript.
> >
> > 3. **Scope of Interventions (Q1):** In the rebuttal, the response regarding interventions remains confined to the additive-noise framework (replacing $\epsilon + \delta^e$ with $\epsilon^e$). This does not address my concern regarding non-additive structural interventions (e.g., changes in the mechanism $f$ or multiplicative/non-linear interventions). Please explain in details how can extend your work to non-additive interventions.
> >
> > 4. **Empirical Validation of Assumptions (Q2):** You noted that Assumption 1 is "difficult to test empirically" in the latent space. This raises a practical concern: if the core assumption cannot be validated on real-world data, how can practitioners determine when the method is safe to apply?
> >
> > 5. **Statistical Significance (Q4):** I find the argument that statistical testing is "inappropriate" or "hard" unconvincing. While deriving theoretical asymptotics for the estimator may be complex, performing standard hypothesis testing (e.g., a paired t-test or Wilcoxon test) on the empirical results (across seeds or test samples) is a standard practice at least for rebuttal period.

---

> > > ### Author Response · Authors · 2025-11-26
> > >
> > > 1. Though we can't say we have understood the paper completely, conceptually, the Ehyaei et al. paper addresses Wasserstein DRO through the lens of structural causal models with a primary focus on individual fairness. Their causal structure concerns how sensitive attributes (like gender, race) causally influence other features, and they define a "Causally Fair Dissimilarity Function" to ensure counterfactual fairness. The DRO framework protects against perturbations that respect this fairness metric.
> > > In contrast, CIRRL focuses on distributional robustness for prediction under multi-environment heterogeneity, where interventions shift the latent causal structure across environments. The goal is robust out-of-distribution prediction, not fairness. Assuming an SCM and aiming for distributional robustness is something they have in common. However, they do two different things. CIRRL handles cases where the relationship between observed covariates X and latent variables Z is nonlinear, and any X variable may depend on any (or **all**) Z. This requires a two-step approach with representation learning. Ehyaei et al. work with observed features directly in their SCM, where the nonlinearity is in the structural equations among observed variables. To achieve the first step, CIRRL relies on recent identifiability results for latent representations (Kivva et al. 2022, Ahuja et al. 2023 or Khemakhem et al. 2022.) under conditions like Gaussian mixture models in the latent space. The work provides a theoretical framework showing that after learning the correct representation, robust prediction in nonlinear settings reduces to robust linear prediction in the latent space, with explicit finite-radius guarantees preserved through this transformation. CIRRL shows optimality over all L^2 functions, not just linear predictors or a restricted function class. This is a stronger guarantee than typical DRO results. Ehyaei et al. consider interventions primarily in the context of counterfactuals for fairness (twins with different sensitive attribute values). Once again, we thank the reviewer for pointing us to relevant literature.
> > >
> > > Instead of conctructing the uncertainty set using a radius in a distribution distance (conservative), CIRRLs uncertainty set is data driven from heterogeneity. It is constructed from the second moments of interventions observed across training environments, with an explicit finite-radius guarantee controlled by γ. The perturbation budget adapts to the observed heterogeneity rather than being pre-specified.
> > >
> > > Explicit robustness radius control: The parameter γ (pre-specified in general) in CIRRL directly controls the trade-off between reference environment performance and robustness to shifts, with Proposition 1 providing an explicit characterization of which perturbations v the model is robust against (those whose second moments are bounded by the weighted combination of training environment second moments).
> > >
> > > 2. We will emphasize this in the revised manuscript.
> > >
> > > 3. Thank you for the further inquiry. Our method is designed around a **linear SCM** in the (unobserved) latent space, which inherently relies on additive noise. This is a careful design choice, as it creates a tractable space for deriving robustness guarantees. Extending the latent SCM itself to be nonlinear (and non-additive noise) would contradict this core premise, as the objective is to learn a representation where relationships are simple and stable. However, it's crucial to note that the **interventions and mechanisms in the observed data can be highly complex and nonlinear**; our nonlinear encoder is tasked with mapping this complexity into the simpler latent structure where our guarantees apply.

---

> > > > ### Author Response · Authors · 2025-11-26
> > > >
> > > > 4. The reviewer raises a critical point regarding the practical application of methods based on untestable assumptions. We agree that this is a fundamental challenge, not just for our method, but for many causal inference and robustness frameworks. Our situation with Assumption 1 is analogous to the standard Instrumental Variable (IV) setting, which **the field widely accepts** as a powerful tool despite similar limitations. In IV analysis:
> > > > i) Only the Relevance condition (that the instrument affects the treatment) is directly testable via the first-stage F-statistic. The F-statistic remains the same using
> > > >
> > > > ii) The core assumptions of Exchangeability and the Exclusion Restriction are not directly testable and must be justified through study design, subject-matter knowledge, and sensitivity analyses.
> > > >
> > > > iii) Similarly, in our setting, practitioners can apply the same rigorous reasoning. The method is "safe to apply" when:
> > > >
> > > > * The theoretical justification for the latent structure is plausible for the specific application domain.
> > > >
> > > > * The method demonstrates robust empirical performance across benchmarks and sensitivity analyses, as we have shown.
> > > >
> > > > We acknowledge that this requires careful consideration, but this does not render the method useless. It places it in the **same category as other principled approaches** that rely on untestable but theoretically-grounded assumptions to provide guarantees against unmeasured confounding and distribution shift.
> > > >
> > > > 5. We apologize for the confusion. We initially thought the reviewer meant something else in their question. We agree with the reviewer that the improvement over DRIG o ICU isnt significant.

---

> ### Author Response · Authors · 2025-11-19
>
> [1] Ehyaei et al. Causal Adversarial Perturbations for Individual Fairness and Robustness in Heterogeneous Data Spaces.
>
> [2] Ehyaei et al. Wasserstein distributionally robust optimization through the lens of structural causal models and individual fairness.
>
> [3] Ehyaei et al. Designing Ambiguity Sets for Distributionally Robust Optimization Using Structural Causal Optimal Transport.
>
> [4] Ahuja et al. Interventional Causal Representation Learning
>
> [5] Khemakhem et al. Variational autoencoders and nonlinear ica: A unifying framework
>
> [6] Guo. Statistical Inference for Maximin Effects: Identifying Stable Associations across Multiple Studies
>
> [7] Londschiem et al. Domain Generalization and Adaptation in Intensive Care with Anchor Regression
>
> [8] von Kuegelgen et al. Representation Learning for Distributional Perturbation Extrapolation

---

### Official Review · Reviewer_fgH7 · 2025-10-31

**Soundness:** 3
**Presentation:** 2
**Contribution:** 3
**Rating:** 6
**Confidence:** 3

**Summary:**

The paper proposes CIRRL (Causality-Inspired Robustness via Representation Learning), a two-step method for achieving distributional robustness in nonlinear settings. The first step learns a nonlinear representation of covariates using a Distributional Principal Autoencoder that maps data to a latent space. The second step applies the DRIG framework on the learned representations to achieve finite-radius robustness guarantees. The key theoretical contribution is establishing that CIRRL achieves optimal worst-case risk among all square-integrable functions under certain assumptions. The method is validated on synthetic and real-world datasets, where CIRRL consistently outperforms existing robust learning methods in worst-case and mean (across environments), and best or second-best in median.

**Strengths:**

- The paper addresses a gap by extending finite-radius robustness from linear to nonlinear settings. This is an important contribution since many/most real-world settings are nonlinear.

- The two-step approach is well-motivated and elegantly combines representation learning with causality-based DRO.

- The theoretical framework is rigorous and establishes the optimality of the learned predictor (Theorem 3).

**Weaknesses:**

**1.** The paper’s presentation, although overall clear, could be improved. The introduction and related work sections are unnecessarily lengthy and could be more focused. The introduction sounds somewhat vague and lacks specificity about the main results and contributions. Ideally, the introduction should be more to-the-point and give a clear, specific (though high-level) summary of the main results. The related work section on the other hand, is overly elaborate. It could be moved to the appendix and condensed in the main paper to make room for an example, or further discussion on the main content.

**2.** The paper lacks concrete examples throughout to ground the abstract concepts.

**3.** The experimental section provides limited insight into when CIRRL's advantages are most pronounced and when existing methods might suffice.

**4.** I apologize if I miss it, but I cannot find a discussion on the choice of hyperparameter $\alpha$ in $L_{RL}$ (Eq. 4). This seems to be important.

**5.** A minor point: the paper is missing a citation to [1], which provides a unifying perspective on causality, IRM, and DRO that would strengthen the paper's positioning. They cite the pieces of work that [1] puts together, but I believe citing [1] itself is appropriate here.

[1] Bühlmann, Peter. "Invariance, causality and robustness." Statistical Science 35.3 (2020): 404-426.

**Questions:**

**1.** Could you provide a concrete example in a tangible scenario and walk the reader through the steps of the argument, the results, the assumptions, and their implications, with the help of that example? Given the overly elaborate intro and related work, I think the paper has room for such an example.

**2.** Can you elaborate on the choice of hyperparameters?

---

> ### Author Response · Authors · 2025-11-19
>
> We thank the reviewer for the constructive suggestions.
> 1) We agree and implemented this exactly. We shortened the introduction substantially; the discussion of related literature has been moved to the appendix with a short pointer in the main paper. Length of related work is due to the fact that the method lives on the intersection of 3 different domains (causality, representation learning, robustness).
> 2) Consider predicting cardiovascular disease risk as an example. The true driver might be arterial inflammation Z, but clinicians do not observe it directly. Instead, they measure indirect markers such like inflamation protein X1, white blood cell count X2, and blood clotting protein X3, all of which reflect Z but also contain environment-specific noise.
> Different hospitals introduce shifts because they use different laboratory equipment and protocols. A model trained at Hospital A may learn patterns tied to A’s high-precision measurements rather than the underlying biology. When deployed at Hospital B with noisier or systematically shifted measurements, its performance fails because those spurious or environment specific correlations no longer hold.
> CIRRL addresses this by learning a representation \hat ϕ(X)^T \hat b that isolates the invariant signal related to inflammation Z and filters out measurement-specific distortions. While the noise patterns differ across hospitals, the causal link between inflammation and disease remains stable, and CIRRL explicitly leverages this invariant structure to remain robust under distribution shifts.
> 3) CIRRL is most useful when (A) the true relationship between observed covariates and the target is nonlinear (so linear methods on X fail), and (B) the distributional shifts act in latent space (additive shifts on Z or environment-conditional latent means), rather than arbitrary large, adversarial changes to the whole X distribution.
> If X to Y is approximately linear (or the true invariant directions are already visible in X), then DRIG or linear baselines may suffice. If X is a nonlinear measurement of a low-dimensional latent Z (e.g., ICU signals, single-cell expression) and shifts occur through changes in latent means/covariances, CIRRL acts by (i) recovering latent structure and (ii) applying finite-radius robustness in that space.
> IRM / invariance methods aim at arbitrarily large shifts and therefore can over-regularize for the finite-shift regime; V-REx/ARM also target different objectives — our experiments (Table 1 & Fig.1) illustrate these regimes.
> 4) Hyperparameter α weights the DPA/mixture prior term that pulls the learned latents toward an environment-conditioned Gaussian mixture. Intuitively, for small α encoder focuses more on reconstruction and predictive performance; latent clustering by environment is weak. For α large, the encoder overfits to the prior and may distort predictive features to match the mixture. Our practical guidance (Appendix D): we typically set
> α∈[10^−4,10^−1]; in our experiments we used α=10^−1.
> We have Figure 7 (Appendix) showing test and worst-case MSE as α varies; performance is stable for small values. α is primarily regularizing toward recognizing distinct environments and toward the mixture assumption used in identifiability; empirically it acts as a helpful but low-weight prior.
> 5) We thank the reviewer for pointing us to futher existing work. We add the citation to the invariance/causality viewpoint. Additional practical clarifications (helpful for reproducing our results), model architecture & tuning (Appendix D): we used 2 hidden layers of width 400 for encoders/decoders, Adam with lr 10^−4, 1000 epochs; these hyperparameters werent tuned.
> Latent dimension k is chosen by visual inspection, using an “elbow”-point in reconstruction loss (Figure 5). Note that over-estimating k is harmless since extra dimensions receive near zero weight in the linear head (see Appendix D).
> The robustness parameter γ should be chosen to match the anticipated magnitude of distributional shift, quantified by moments of shifts (cf. the uncertainty set, line 310). In the absence of test data, γ must be chosen using domain knowledge, this is consistent with DRO practice of specifying a robustness radius a priori.
> Appendix D contains an analysis. Practically, overspecification does not cause significant decline in performance.
> We Thank the Reviewer again for the detailed review.

---

### Official Review · Reviewer_5FbT · 2025-11-01

**Soundness:** 3
**Presentation:** 3
**Contribution:** 3
**Rating:** 4
**Confidence:** 4

**Summary:**

This paper proposes a novel framework, CIRRL (Causality-Inspired Robustness via Representation Learning), which aims to achieve distributional robustness in nonlinear settings. The method integrates recent advances in identifiable representation learning with causality-inspired robust optimization. Specifically, the authors first learn a nonlinear latent representation that is affine-identifiable across environments, and then apply a DRIG-style finite-radius robust regression in this latent space. The paper provides theoretical guarantees for robustness and identifiability under a structural causal model (SCM) formulation and demonstrates empirical superiority on both synthetic and real-world biomedical datasets.

**Strengths:**

•	Clear writing and presentation: The paper is well-written and easy to follow. The motivation, intuition, and mathematical formulation of CIRRL are presented clearly, and the connections with prior works such as DRO, DRIG, and IRM are well articulated.
•	Conceptual novelty: By combining representation learning with causality-based robustness, CIRRL offers a practical way to handle nonlinear dependencies and additive perturbations. This is a meaningful step toward bridging causality and distributional robustness.
•	Empirical performance: The experiments are convincing and demonstrate that CIRRL achieves consistently strong robustness compared with existing baselines (ERM, IRM, V-REx, ARM, DRIG). The proposed approach seems practical and potentially applicable to a wide range of real-world scenarios.

**Weaknesses:**

•	Limited discussion on causal mechanisms. While the method is labeled “causality-inspired,” the paper does not provide a concrete definition or interpretation of the causal mechanisms involved. The SCM formulation is mainly used to justify additive perturbations, but there is little insight into mechanism-level invariance or identifiability beyond affine transformations.
•	Affine representation assumption may not generalize to complex data (e.g., images). The proposed affine identifiability assumption is questionable for high-dimensional or convolutional data. The paper claims general applicability (including to image data), yet no experiments are conducted on such domains. It is unclear whether the linear latent assumption remains valid when spatial and hierarchical dependencies are dominant.
•	Theory is internally consistent but relies on tricky assumptions. The theoretical results only show optimality of the estimator under the self-defined loss function and a series of strong assumptions (existence of a linearizing nonlinear map, elliptical noise, Gaussian mixture latent structure). These make the results mathematically elegant but potentially fragile and hard to verify empirically.
•	Empirical validation does not directly test competing uncertainty-set definitions. Although the paper emphasizes CIRRL’s advantage in modeling nonlinear relationships and finite-radius robustness, the experiments do not include direct comparisons with other uncertainty-set formulations (e.g., Wasserstein-based DRO, moment-based sets). This weakens the claim of superiority in modeling distributional uncertainty.

**Questions:**

1.	In Equations (2) and (3), the additive interventions δ_e and v are used to model distributional shifts. Could the authors clarify how these additive perturbations capture realistic SCM-level distribution shifts beyond mean or variance changes?
2.	How should the dimension of the latent representation Z be chosen relative to X? Is it a hyperparameter, or are there theoretical or empirical guidelines?
3.	Since CIRRL relies on a linear regression in the latent space, do the learned linear coefficients b offer any interpretability benefits? Or does the affine transformation φ(·) essentially make the model another black-box representation?

---

> ### Author Response · Authors · 2025-11-19
>
> We thank the reviewer for the careful and constructive feedback.
> 1. Causality is **used for the theoretical guarantees**, not as part of the algorithm itself.
> Our method does not perform causal discovery or mechanism recovery; instead, it uses the SCM formalism to define the uncertainty set (latent interventions) and to derive the **finite-radius robustness guarantee**.
> Algorithmically, CIRRL consists simply of
> (i) learning a nonlinear latent representation via a Distributional Principal Autoencoder, and
> (ii) applying a DRIG-style finite-radius regression in that space.
> Thus causality enters the *mathematical justification* rather than the optimization steps. We added explicit text in Secs. 3.1–3.3 explaining this point.
> 2. Affine identifiability and images
> Identifiability is **not an arbitrary assumption**; it is a property that holds under explicit, well-studied conditions (e.g., GMM or piecewise-affine decoders, see Lemma 2).
> We revised the wording to emphasize that these are *conditions for theoretical identifiability*, not ad-hoc modeling choices.
> Images: we agree and correct the claim of general applicability.
> Many image problems are anti-causal or have spatial structure where a linearizing latent may not exist.
> CIRRL is intended for settings where such a latent representation is plausible. We explicitly acknowledge this limitation in the revision.
> 3.  We appreciate the comment and clarify the role of each assumption.
> * The existence of a nonlinear map that linearizes the SCM defines the model class under which guarantees hold, it is not a technical assumption.
> * The Gaussian-mixture/piecewise-affine conditions are (one of many) sufficient conditions for affine identifiability.
> * The elliptical-noise assumption is used only for the optimality proof in Theorem 3.
> To demonstrate robustness beyond these conditions, we conduct a misspecification experiments with non-Gaussian (χ²) shifts; CIRRL remains stable and superior to baselines.
> We also add **Theorem 4**, which gives an explicit bound showing that if the learned latent is δ-close to the true one, the excess error grows only linearly in δ. Hence even imperfect representations remain near-optimal. While we agree that such assumptions are hard to verify empirically, this is true for nearly all identifiable-representation frameworks. Our contribution is to make them explicit, show empirical robustness when violated, and quantify the resulting error.
> 4. We acknowledge this point and expanded the explanation in Sec. 4.
> Our finite-radius, causality-based uncertainty set is fundamentally different from Wasserstein or moment-based DRO. A direct comparison would require arbitrary calibration of radii or transport costs, which risks being misleading. Instead, we follow the **DRIG/anchor-regression** literature, which already demonstrates superiority over classical DRO on similar datasets.  We now cite those results explicitly and clarify why DRIG is the correct baseline for our threat model.
> (If it deems essential, we are willing to add a small appendix experiment comparing to a Wasserstein-DRO variant.)
>
> * Q1. Additive interventions δₑ and v
> The interventions are additive in the *latent* SCM, not necessarily in observed X.
> Because ϕ can be highly nonlinear, additive perturbations in Z produce complex nonlinear shifts in X.
> Our uncertainty set constrains only the first two moments of v (elliptical family); this assumption is used only in Theorem 3. In practice, δₑ can also for example, be skewed, as confirmed in the experiments.
> * 2. Choice of latent dimension k hyperparameter
> We recommend an **elbow-plot** of reconstruction or validation loss (Appendix D), or slight over-specification, extra dimensions simply get near-zero weights. Visual inspection or cross-validation across environments works well in practice.
> * 3. Interpretability of linear coefficients b
> If the learned representation differs from the true one by an invertible matrix M (i.e., $\hat{z} = Mz $, then
> $ \hat b = M^{-1} b $ and predictions satisfy $ \hat{y} = \hat b^{\top}\hat z = b^{\top}z$.
> Thus the predictions are identical, but individual coefficients lose direct interpretability unless M is known.
> We added this algebraic explanation to Sec. 3.2–3.3. We thank Reviewer 5FbT again for the detailed feedback and hope these clarifications address all outstanding issues.

---

### Meta-Review · Area_Chair_bTGJ · 2026-01-11

**Summary:**

There was a agreement among reviewers that the paper tackles an important and timely problem, extending causality-inspired finite-radius robustness guarantees from linear to nonlinear models, and does so with a technically sophisticated and well-motivated framework. However several concerns limit enthusiasm for acceptance such as i) theoretical guarantees rely on a stack of strong assumptions (latent linear SCM, affine identifiability via GMM or piecewise-affine decoders, elliptical noise, and ii) novelty relative to recent causality-inspired DRO and SCM-based robustness, iii) limited insight into when proposed method improves over DRIG, mostly lacking statistical significance testing on small real datasets.

**Reviewer Concerns:**

The authors did a descent job in clearly explaining  that causality is used for theoretical justification, not causal discovery or mechanism recovery, which addresses confusion raised by multiple reviewers. Claims about general applicability (e.g., to images) are explicitly corrected, and related work omissions are fixed, directly addressing concerns from Reviewers fgH7 and qm5p. The clarity improved as authors made introduction and related work shorter and moved to the appendix; concrete examples are added; hyperparameter choices are clarified.

**Reviewer Scores:**

Although the authors clarify conceptual differences with prior work., reviewer qm5p remains unconvinced that the distinction and added value are fully compelling. The late acknowledgment that some improvements (e.g., over DRIG on ICU) are not statistically significant reinforces earlier doubts rather than resolving them and needs to be taken care of more rigorously in subsequent versions.

---

### Decision · Program_Chairs · 2026-01-26

Reject